# Crystal structure of a highly conserved enteroviral 5′ cloverleaf RNA replication element

Naba K. Das [1], Nele M. Hollmann[1,2], Jeff Vogt[1], Spiridon E. Sevdalis[3], Hasan A. Banna [1], Manju Ojha[1] & Deepak Koirala [1] ✉

The extreme 5′-end of the enterovirus RNA genome contains a conserved cloverleaf-like domain that recruits 3CD and PCBP proteins required for initiating genome replication. Here, we report the crystal structure at 1.9 Å resolution of this domain from the CVB3 genome in complex with an antibody chaperone. The RNA folds into an antiparallel H-type four-way junction comprising four subdomains with co-axially stacked sA-sD and sB-sC helices. Long-range interactions between a conserved A40 in the sC-loop and Py-Py helix within the sD subdomain organize near-parallel orientations of the sA-sB and sC-sD helices. Our NMR studies confirm that these long-range interactions occur in solution and without the chaperone. The phylogenetic analyses indicate that our crystal structure represents a conserved architecture of enteroviral cloverleaf-like domains, including the A40 and Py-Py interactions. The protein binding studies further suggest that the H-shape architecture provides a ready-made platform to recruit 3CD and PCBP2 for viral replication.

The *enterovirus* genus of the *Picornaviridae* family includes numerous pathogenic viruses responsible for many human diseases, such as the common cold, poliomyelitis, acute flaccid paralysis, and myocarditis[1–3]. These viruses contain a (+)-sense single-stranded RNA genome with about 7500 nucleotides (nts), which is polyadenylated at the 3′-end and linked covalently with viral protein VPg at the 5′-end (Fig. 1a)[4–7]. The entire genome consists of a single open reading frame (ORF) flanked by the highly conserved 5′ and 3′ untranslated regions (UTRs). The ~750-nt 5′-UTR harbors modular RNA domains essential for the viral genome translation and replication within the host cells (Supplementary Fig. 1). Approximately 660 nts of the 5′-UTR from position 90 to 750, located immediately upstream of the ORF, contribute to an internal ribosome entry site (IRES) that promotes viral genome translation through a cap-independent mechanism[8–11]. The remaining 90 nts at the extreme 5′-end of an enteroviral genome are essential for replication and have been proposed to adopt a cloverleaf-like RNA secondary structure (5′CL) that serves as a platform to integrate viral and cellular protein factors required for initiating the viral

genome replication[8,12–15]. Here, we report a high-resolution crystal structure of an intact enteroviral cloverleaf RNA from a member of enterovirus B species – the coxsackievirus B3 (CVB3).

The 5′CL is highly conserved among all members of the *enterovirus* genus. Because of such high conservation of the structural features among enteroviral 5′CLs, the chimeric enteroviral genomes with swapped 5′CLs have been shown to generate viable virus particles[16–18]. The proposed RNA secondary structure consists of four highly organized subdomains designated sA, sB, sC, and sD (Fig. 1a). The subdomain sA forms the base stem of the cloverleaf, whereas the subdomains sB, sC, and sD each fold into a distinct stem-loop structure. The subdomain sD recruits the viral fusion protein 3CD – a precursor of the viral protease 3C and the viral RNA-dependent RNA polymerase (RdRp) D – through specific interactions of the sD loop with 3C protease[17,19–21]. The subdomain sB with a C-rich sequence in its loop recruits the host poly(C)-binding protein (PCBP) to facilitate the circularization of the viral genome through interactions with poly(A)-binding protein (PABP) complexed with the 3′-end poly(A) tail[19,22–29].

[1]Department of Chemistry and Biochemistry, University of Maryland Baltimore County, Baltimore, MD 21250, USA. [2]Howard Hughes Medical Institute, University of Maryland Baltimore County, Baltimore, MD 21250, USA. [3]Department of Biochemistry and Molecular Biology, University of Maryland School of Medicine, Baltimore, MD 21201, USA. ✉e-mail: dkoirala@umbc.edu

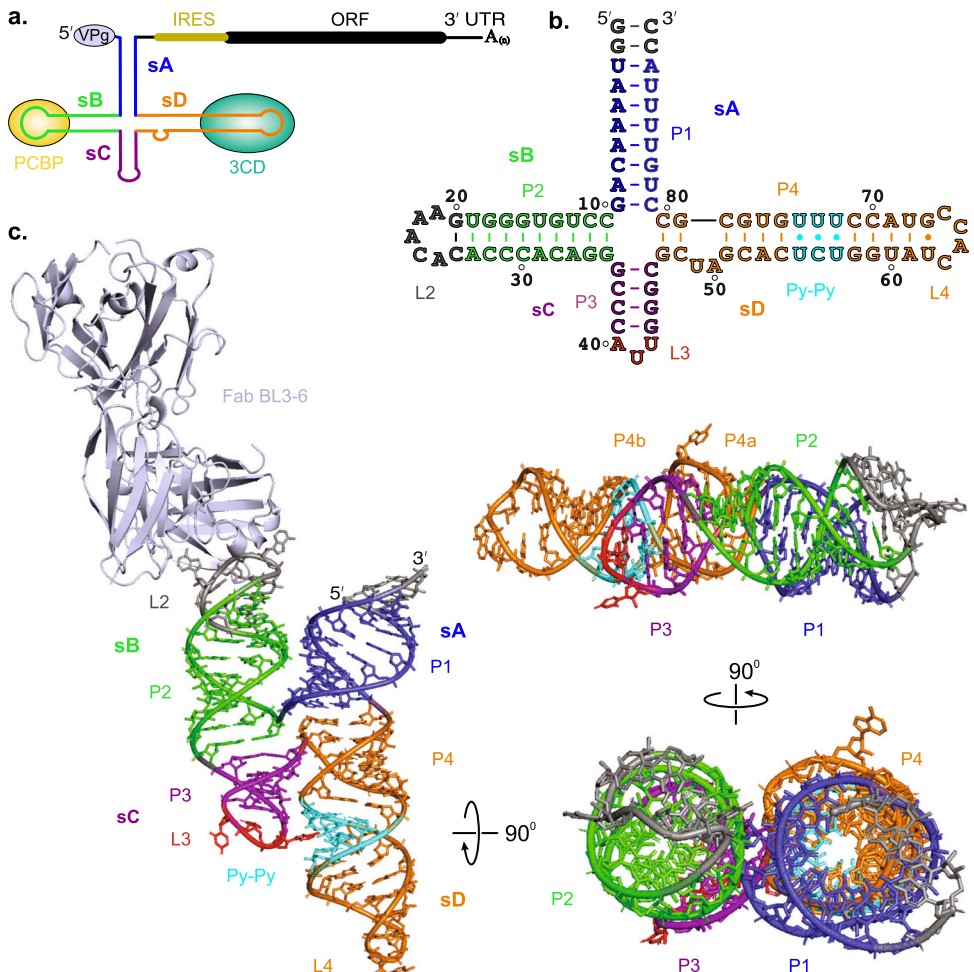

**Fig. 1 | Enteroviral genome organization and overall structure of CVB3 5′CL in complex with Fab BL3-6. a** Schematic of an enterovirus genome depicting the location of the 5′CL and its proposed secondary structure with putative binding sites for viral 3CD and host PCBP proteins. **b** Secondary structure of CVB3 5′CL2a crystallization construct based on previous biochemical analysis, where the Fab BL3-6 binding epitope 5′-GAAACAC-3′ motif replaces the L2 loop of the subdomain sB. The nucleotides colored in gray represent the mutations or insertions compared to the wild-type sequence. **c** The crystal structure of the CVB3 5′CL2a cocrystallized with the Fab BL3-6 and solved at 1.9 Å resolution. The Fab is obscured in rotated views of the RNA structure for clarity. Figure panels and corresponding labels are colored analogously for facile comparison.

A stem-loop RNA domain, *cre*, within the 2C coding region of the viral genome also interacts with the 3CD[30–32], promoting the uridylation of the viral VPg protein[31,33], which afterward serves as a primer for the (-)-strand RNA synthesis by the RdRp D[30,34]. Additionally, the sequence and structural integrity of the 5′CLs have also been shown to influence the VPg uridylation and genomic stability, underscoring multiple critical roles of RNA structural features within the enteroviral 5′CLs[14,35]. Despite intense selection pressures, the high conservation of the 5′CL in enteroviral genomes also highlights the viral requirements to preserve primary, secondary, and tertiary RNA structures of the 5′CL for interactions with the viral-derived and host proteins during the viral genome replication. However, we lack the high-resolution, three-dimensional structures of intact enteroviral 5′CL, constraining our understanding of this fundamental virological process that has tremendous potential for developing targeted therapeutics against enteroviral infections.

Several biochemical and biophysical methods have been used to study the structures of enteroviral 5′CL subdomains and their molecular interactions with PCBP and 3C or 3CD proteins[36–44]. Earlier studies focused on understanding isolated sD structures using nuclear magnetic resonance (NMR) spectroscopy, which included the NMR-based structures of sD subdomains from CVB3, rhinovirus B14 (RVB14), and an enteroviral consensus sequence[36–38,40,43]. Consistent with the

chemical probing and SHAPE (selective 2′ hydroxyl acylation analyzed by primer extension) analyses in the context of the full-length 5′CLs or the entire 5′-UTRs[39,42,45], the NMR structures showed a well-conserved structure of the sD variants that adopts a hairpin-like architecture with an A-form helical stem capped by a tetraloop. Interestingly, the structures also revealed a unique pyrimidine-pyrimidine (Py-Py) base-paired region within the sD stem, which was previously predicted to form a 3 × 3-nt symmetric bulge. The sequence and the structure of this region are well-conserved among all enteroviruses, and as shown previously for CVB3, deletion of a nucleotide, but not the compensatory point mutation, in this region reduces the sD interactions with the 3C protease, suggesting that the integrity of the structural features formed within Py-Py region is important for the 3C binding[14,20]. However, the NMR data did not support the direct interactions between this region and the 3C protease. The direct interactions with the protein only involved the sD loop nucleotides and two G-C base-pairs flanking the Py-Py region[38]. Later, Warden et al. reported an NMR structure of an isolated 24-nt sB sequence from the RVB14, which folds into a hairpin with a predominantly A-form helical stem capped by a disordered 8-nt loop[41]. The C-rich region within the loop, the recognition site for the host PCPB, was exposed to solvent, consistent with the earlier biochemical studies[41]. The stem region had an accessible major groove, indicating a potential site for the protein interactions.

Most recently, Warden et al. derived the structural models of the full-length RVB14 5′CL using NMR and small-angle X-ray scattering (SAXS) techniques[43]. The proposed structural models adopted two conformations depending upon the presence and absence of $Mg^{2+}$ in the solution. In the absence of $Mg^{2+}$, the 5′CL assumed an extended conformation similar to a *cK*-type four-way junction structure with the sB and sD subdomains residing almost perpendicular to one another, whereas in the presence of $Mg^{2+}$, it assumed a more compact conformation juxtaposing these sB and sD subdomains[43]. Nevertheless, without the high-resolution structures of other subdomains (sA and sC) and an intact 5′CL, the topological arrangements of the 5′CL subdomains and the structural features that facilitate their binding interactions with the viral 3C, 3CD, and host PCPB proteins remains largely unknown.

Here, we use a synthetic antibody fragment (Fab) as a chaperone to crystallize and determine the 1.9 Å resolution crystal structure of the full-length 5′CL from CVB3. The RNA folds into an H-type four-way junction architecture with no unpaired nucleotides within the junction. The subdomains assemble into two sets of coaxially stacked helices, with each coaxial stacking forming almost a continuous A-form helix. Within the four-way junction, the sA helix stacks on the sD helix and the sB helix on the sC helix. The crystal-derived secondary structure agrees well with those derived previously from biochemical probing, SHAPE, and the NMR studies of isolated subdomains from various 5′CLs. Surprisingly, our crystal structure revealed an unprecedented tertiary interaction between the highly conserved sB loop and the sD helix Py-Py region. These interactions agree well with our solution NMR results, indicating that the structural features observed in our crystal structure represent that in the solution. The unique arrangement of these structural features also suggests that they are important for stabilizing the overall structure of the 5′CL that perfectly positions its sB and sD subdomains for interactions with 3C or 3CD and PCBP proteins, which agrees with our binding studies using recombinant CVB3 3C protease and human PCBP2 using isothermal titration calorimetry (ITC) and gel electrophoresis. Our high-resolution structural determination of an intact 5′CL from an enterovirus will facilitate the efforts to understand the structural organization and functional roles of the 5′CL during enteroviral genome replication. Moreover, given a high degree of sequence and secondary structure conservation of 5′CLs among enteroviruses, this study will assist in understanding the 5′CL-mediated enteroviral replication mechanism, which has tremendous potential for developing targeted therapeutics for viral replication inhibition, potentially leading to more effective treatment of many human diseases caused by the enteroviral infections.

## Results

### CVB3 5′CL crystallization constructs

The CVB3 5′-UTR contains seven modular domains comprising highly organized RNA secondary structures (Supplementary Fig. 1)[39]. The domains designated II to VII include the IRES elements responsible for viral genome translation via a cap-independent mechanism, whereas domain I represents the 5′CL structure[8–15]. The 90-nt wild-type (WT) crystallization construct from CVB3 isolate 28 encompasses the entire 5′CL (nts 2 to 88) with two additional G-C pairs at the beginning of the sA helix (Supplementary Fig. 2). Our efforts to crystallize this WT 5′CL construct were unsuccessful; therefore, we followed a chaperone-assisted RNA crystallization strategy. We have been developing and employing Fab fragments as chaperones for Fab-RNA complex crystallization and as molecular replacement models for initial phasing during the structure determination process[46–51]. One of the approaches in this strategy utilizes a Fab-epitope module, which involves grafting the RNA epitope into the target RNA to enable its complex formation with the Fab, allowing subsequent crystallization of the Fab-RNA complex[46–51]. Among others, Fab BL3-6 has been a very effective chaperone to crystallize and determine the high-resolution crystal

structures of several RNA targets[46,47,49,50,52,53]. The Fab binds to a hairpin RNA with 5′-AAACA-3′ pentaloop sequence closed preferably by a G-C pair, which allows facile engineering of the epitope sequence into the target RNA[46,47,49,50,52,53].

Based on the putative secondary structures of the 5′CL derived from biochemical probing, SHAPE, and NMR studies on intact 5′CL and its isolated domains, we prepared three separate RNA constructs, 5′CL2, 5′CL3, and 5′CL4, for crystallization by replacing the L2, L3, and L4 loops of the WT 5′CL with the 5′-GAAACAC-3′ sequence to create a binding site for the Fab BL3-6 in each RNA construct (Supplementary Fig. 2). The native polyacrylamide gel electrophoresis assays confirm that Fab BL3-6 binds explicitly to these three RNA constructs (Supplementary Fig. 3), consistent with previous reports for other RNA constructs grafted with this Fab BL3-6 binding sequence. Because Fab BL3-6 binds to its epitope sequence only as a hairpin RNA, these observations also indicated that each 5′CL subdomain, sB, sC, and sD, likely adopts distinct stem-loop structures, consistent with previous biochemical and NMR studies.

### Crystallization and structure determination of CVB3 5′CL

Following the binding tests, we set up the crystallization trials for all three RNA constructs, 5′CL2, 5′CL3, and 5′CL4, in complex with Fab BL3-6. We observed crystals for 5′CL2 and 5′CL4 complexes but not for the 5′CL3 complex. Out of the 480 conditions screened for each complex, the 5′CL2 complex crystallized in fewer conditions (2) than 5′CL4 (16). The analogous trials for both constructs without Fab BL3-6 did not produce any crystals, suggesting a substantial role of the Fab in facilitating the 5′CL2 and 5′CL4 crystallization. Unfortunately, the crystals for these complexes diffracted to ~8 Å resolution only, which was inadequate for a high-resolution structure determination. To obtain better quality crystals and a high-resolution diffraction dataset, we prepared several 5′CL2 mutant constructs to set up the crystallization trials in complex with Fab BL3-6. For each L2 loop mutant, we predicted the sD hairpin models using ROSIE[54,55], with particular attention to the L2 tetraloop conformation. The models predict that the G66C mutant adopts a GNRA-type tetraloop conformation compared to a UNCG-type tetraloop for the wild-type L2 (Supplementary Fig. 4). As the GNRA-type tetraloops are known to facilitate crystal contacts[56], we used this G66C mutation in the context of the 5′CL2, designated 5′CL2a (Fig. 1b), which yielded robust crystals in complex with the Fab that diffracted to 1.9 Å resolution. For the subsequent structure-solving process with this data, we used a previous crystal structure of Fab BL3-6 (PDB code: 6B14)[46] as a molecular replacement model and readily obtained the initial phases. After the initial phasing, the high-resolution electron density map allowed us to model the RNA nucleotides unambiguously. The structure was then solved through iterative rounds of model rebuilding and refinement at 1.9 Å resolution, and the final model of the 5′CL2a-BL3-6 complex converged with $R_{work} = 20.9\%$ and $R_{free} = 25.9\%$. The data collection and refinement statistics details are shown in Supplementary Table 1. The final structural model of the 5′CL2a-BL3-6 complex with the $2|F_o|\text{-}|F_c|$ electron density map and that colored according to the crystallographic B-factors are shown in Supplementary Figs. 5 and 6, respectively.

### The overall structure of the CVB3 5′CL

The crystallographic asymmetric unit ($a = 122.20$, $b = 48.70$, $c = 144.65$, $\alpha = 90$, $\beta = 112.92$, and $\gamma = 90$) with a lattice space group C 1 2 1 contains a single Fab-RNA complex. Analysis of the interfacial area within the unit cell using PDBePISA[57] revealed that including the Fab-RNA binding interface, the Fab-Fab, Fab-RNA, and RNA-RNA crystal contacts account for ~142, 2993, and 266 Å$^2$ area, respectively. While Fab makes the majority of crystal contacts (~92% including the Fab-RNA binding interface), the RNA-RNA interactions through coaxial stacking of symmetry-related sA helices within the crystal lattice (Supplementary Fig. 7) account for ~8% of the interfacial area. The Fab-RNA contacts in

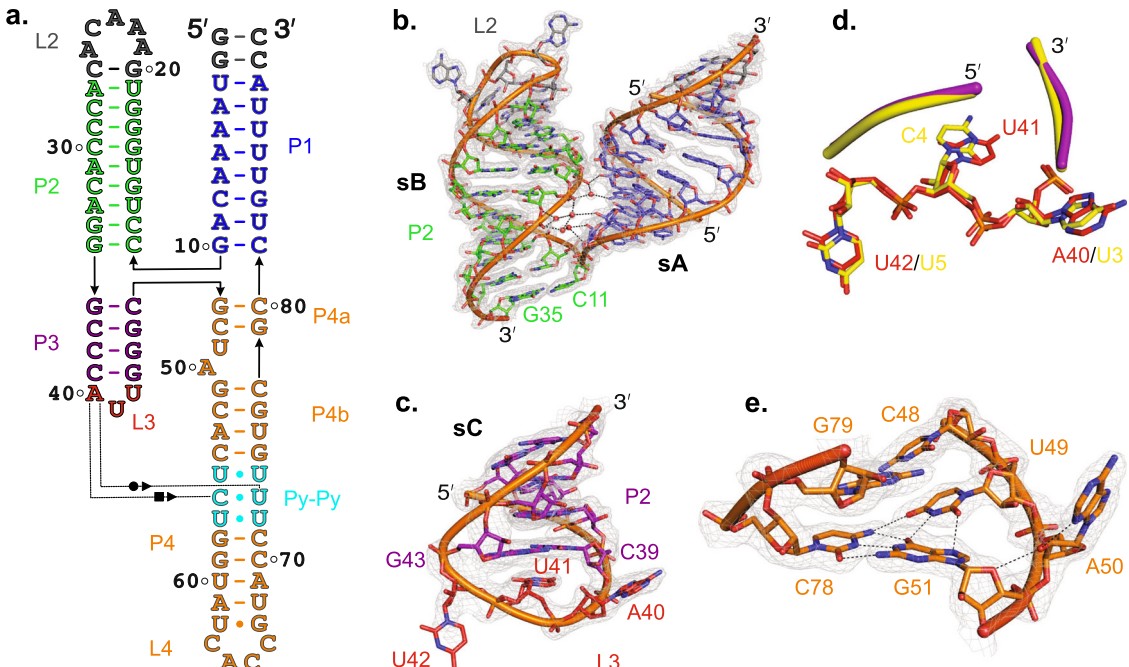

**Fig. 2 | Structural features of the CVB3 5'CL subdomains. a** The crystal-derived secondary structure of the CVB3 5'CL2a construct. The dotted line indicates tertiary interactions between the L3 loop A40 and the P4 helix Py-Py region. **b** The tertiary interactions between the sA and sB subdomains. **c** Structure of the trinucleotide loop within the sC subdomain. **d** Superposition of the 5'-ccAUUgg-3' and the 5'-ccUCUgg-3' stem-loops (RMSD = 0.44 Å) from the crystal structures of CVB3 5'CL2a (red triloop with purple stem) and *Agrobacterium* tRNA[Leu] (yellow, PDB code: 5AH5)[60]. Only the loop nucleotides are shown in detail for clarity. **e** Structure of the dinucleotide bulge within the sD subdomain P4 helix. Black dashed lines and red spheres in (**b**) and (**e**) reflect heteroatoms within hydrogen bonding distances and water molecules, respectively. Gray mesh represents the $2|F_o| - |F_c|$ electron density map at contour level 1σ and carve radius 1.8 Å.

the crystal lattice other than the binding interface mainly involve electrostatic interactions of the

RNA backbone with lysine and arginine side chains from the heavy and light chains of the symmetry-related Fabs, indicating that Fab facilitated crystal packing by neutralizing the negative charges on the RNA surface (see Supplementary Fig. 8 for details).

The crystal structure of the 90-nucleotide 5'CL2a construct assumes a compact, H-type antiparallel four-way junction architecture composed of one stem and three stem-loop regions (Fig. 1c). The four helical stems composing the four-way junction designated P1, P2, P3, and P4 assemble into two sets of coaxially stacked helices, where P1 stacks on P4 and P2 stacks on P3. Within the four-way junction, with no unpaired nucleotides and perfect cross-strand stacking of the P1 helix G10 with P4 helix G47 and P2 helix C11 with P3 helix C46, each set of these coaxial stacking forms almost a continuous A-form helix. As expected, the L2 loop closes the P2 helix and binds the Fab BL3-6, consistent with the Fab-loop interactions observed for other RNA-BL3-6 complex crystal structures[46,47,49,50,52,53]. Similarly, a trinucleotide loop L3 and a tetranucleotide loop L4 close the P3 and P4 helices, respectively. A dinucleotide bulge with U49 and A50 separates the helix P4 into sub-helices, P4a and P4b, where the U49 remains within the continuous helical stack, and A50 flips out of the helix to interact with a symmetry mate Fab molecule in the crystal lattice (Supplementary Fig. 8). The P4b helix contains a Py-Py region with a U•U base-pair on either side of the central C•U base-pair.

The crystal-derived secondary structure of the 5'CL2a (Fig. 2a) in terms of base-paired helices (P1, P2, P3, and P4), dinucleotide bulge, Py-Py base-paired region, and loops (L2, L3, and L4) is well consistent with the previously proposed CVB3 5'CL secondary structural model in the context of complete 5'-UTR based on SHAPE reactivity analysis and biochemical probing with dimethyl sulfate (DMS), 1-Cyclohexyl-3-(2-morpholinoethyl) carbodiimide metho-p-toluenesulfonate (CMCT) and Ketoxal (Fig. 1b, see Supplementary Fig. 9 for details)[39,42,45].

Furthermore, the structures of the entire sD subdomain and the sB helix are almost identical to those determined previously in isolation by solution NMR approaches (Supplementary Fig. 9)[37,38,40,43], suggesting that our 5'CL2a crystal structure represents its solution conformation.

## Structural features of the sA and sB subdomains

In the crystal structure of 5'CL2a, the first ten nucleotides (G1–G10, 5'-end) form canonical Watson-Crick base-pairs with the last ten nucleotides (C81–C90, 3'-end) to constitute the P1 helix, representing the entire sA subdomain (Fig. 2a, b). Within the P1–P2 junction, a sharp turn between G10 and C11 occurs, bending the entire P2 helix to position it almost parallel to the P1 helix (Fig. 2a, b). The P2 helix is then capped by the loop L2—the binding site for the Fab BL3-6, forming the complete sB subdomain structure (Fig. 2a, b). In the WT 5'CL, the L2 loop is expected to form an eight-nucleotide C-rich motif that constitutes the binding site for the PCBP (Supplementary Fig. 2). Excluding the closing G20-C26 base-pair of the BL3-6 binding motif, the P2 helix is comprised of nine Watson-Crick base-pairs, forming a typical A-form helix (Fig. 2b), which is consistent with the previous biochemical probing and the NMR models of RVB14 sB subdomain[41], indicating that this sB helix structure is well conserved among enteroviruses. Moreover, we observed tertiary contacts between the backbones of P1 and P2 helices through direct and water-mediated hydrogen bonding interactions, stabilizing the overall juxtapositions of the sA and sB (Fig. 2b). Specifically, G83 2'-OH forms a direct hydrogen bond with the C29 backbone phosphate oxygen. Nevertheless, we did not observe a significant deviation of the P2 helix from a typical A-form helix, including a widening of the major groove, as proposed previously based on the NMR structures of the sB subdomain from RVB14[41]. Although it is possible that the differences in major groove widths for CVB3 and RVB14 sB helix reflect the differences in methodologies used for these RNA structure determination[58], the structural differences

between our CVB3 and RVB14 sB helices are likely because of the different number of base-pairs flanking the 4-way junction that constitute the sB helix (Supplementary Fig. 10). The nine base-paired sB helix in the CBV3 sB subdomain forms a near complete helical turn compared to a slightly over a half helical turn formed by the seven base-paired sB helix in the RVB14. Additionally, compared to all canonical base-pairs in the CVB3 sB, the RVB14 sB helix contains a non-canonical G•U pair (Supplementary Fig. 10).

## Structural features of the sC subdomain

The four canonical Watson-Crick base-pairs involving the nucleotides G36–C39 and G43–C46 form the P3 helix (Fig. 2a). Within the P2-P3 junction, the G36-C46 of the P3 helix stacks coaxially on the G34-C11 of the P2 helix, forming almost a continuous A-form helix (Figs. 1c, 2a). A trinucleotide loop L3 with the 5′-AUU-3′ sequence closes the P3 helix, consistent with the previous biochemical probing and SHAPE results[39,42,45]. Within the loop (Fig. 2c), a sharp turn in the helix between C39 and U41 projects the nucleotide A40 well out of the helical axis, allowing its tertiary interactions with the P4 helix. The nucleotide U41 with C2′-endo *syn* glycosidic conformation stacks on the loop-closing C39-G43 base-pair making it less accessible for modifications. The second sharp turn between U41 and G43 flips the nucleotide U42 out of the helical axis, potentially making crystal contacts with a symmetry-mate Fab in the crystal lattice. Nevertheless, the poor electron density observed for the U42 nucleobase with a high B-factor suggests its highly dynamic nature. These structural features observed for this triloop match the previous biochemical probing results in the solution that the U41 and U42 nucleobases in this triloop were moderately and strongly, respectively, susceptible to modification by CMCT[39,45]. Moreover, although triloops are less common than tetraloops in RNA structures, searching this triloop motif in previously reported RNA structures using the RNA CoSSMos database[59] resulted in no-hit for an exact 5′-AUU-3′ sequence in the triloops. However, the structure of this 5′-ccAUUgg-3′ triloop was superimposable with a 5′-ccUCUgg-3′ triloop (Fig. 2d, RMSD = 0.44 Å) observed previously in the crystal structure of *Agrobacterium* tRNA^Leu (PDB code: 5AH5)[60], suggesting that these triloops represent a distinct class of RNA motifs.

## Structural features of the sD subdomain

The C46 and G47 nucleotides make a sharp turn within the P3–P4 junction and bend the entire P4 helix, which positions it almost parallel to the P3 helix and allows coaxial stacking with the P1 helix (Figs. 1c, 2a). Next to the junction, the G47 and C48 base-pairs with C80 and G79, respectively, to form the P4a helix. An asymmetric dinucleotide bugle with U49 and A50 separates the P4a from the P4b helix, where U49 remains within the helical stack and involves in the U49•G51-C78 base-triple formation (Fig. 2a, e). While G51 and C78 form Watson-Crick base-pairs, U49 and G51 interact through the Watson-Crick and Hoogsteen hydrogen bonding. Additionally, U49 O4 makes a hydrogen bond with the C78 amino group. The nucleotide A50 flips out of the helix and interacts with a symmetry-mate Fab molecule in the crystal lattice, which perhaps is crucial for crystallization (Supplementary Fig. 8). Also, the 2′OH of the A50 is within the hydrogen bonding distance of its N3 and the G51 O4′. Consistent with the high conservation of the bulge nucleotides, these structural features seem essential for the stability of the sD helix. While the crystal structure may represent a sampled conformation of the bulge with flipped-out A50 stabilized by the crystal contact, a strong hydrogen bonding network within the bulge supports a stable rather than a dynamic configuration of this bulge. Also, the 49th nucleotide is less conserved compared to A50 among enteroviral 5′CLs, indicating that flipped-out A50 may be important for interactions with the 5′CL binding proteins. The bulge perhaps helps preserve a correct positioning of the Py-Py region for tertiary interactions with the sC loop without perturbing the helical axis of the sD subdomain. Following the bulge, the nucleotides

C52–A61 form Watson-Crick base-pairs with the U68–G77, including the highly conserved Py-Py base-pairs. In the Py-Py region, nucleotides U55, C56, and U57 base-pairs with U74, U73, and U72 via non-canonical Watson-Crick hydrogen bonds (Fig. 3a). Interestingly, the Py-Py double helix is slightly narrower (phosphate to phosphate diameter = 16.8 ± 0.3 Å, Fig. 3b) compared to the double helices flanking the region (phosphate to phosphate diameter = 18.6 ± 0.6 Å, Fig. 3b), which is consistent with previous observations in NMR models of CVB3 and RVB14 sDs[36–38,40,43]. However, these unusual consecutive Py-Py base-pairings (U55•U74, C56•U73, and U57•U72) created no significant deformation within the P4 from a standard A-form helix or altered the relative positions of the helical ends (see Supplementary Fig. 11 for details). Finally, the loop L4 with a wobble U62•G67 closing base-pair, which assumes a well-defined tetraloop structure (Figs. 1c, 2a), caps the P4 helix.

## Comparison of the crystal and NMR structures of the sD subdomain

To compare the observed structural features of the sD subdomain with previously reported NMR models, we superposed the sD subdomain crystal structure with the previous NMR models (lowest energy structures) of sDs for CVB3 and an enteroviral consensus sequence (Fig. 3c, RMSDs = 1.947 Å, and 3.461 Å, respectively)[37,38]. While we observed some differences in the bulge and loop L4 structures, the structures of the Py-Py helix were very similar (Fig. 3d, RMSDs = 0.430 Å, and 1.266 Å, respectively). Interestingly, in contrast to our crystal structure, the consensus sD structure showed flipped-out U49 and helically stacked A50. This configuration allows the formation of an H-bonding network involving C48, U49, A50 and G77, including a base-triple A50•C48-G77 instead of the U49•G51-C78 observed in the crystal structure (Supplementary Fig. 12). Consistent with this observation, previous studies using cell-free systems have shown that a single nucleotide bulge is sufficient for the (-)-strand synthesis[14,20]. Each configuration may exist, or the overall four-way junction fold of the CVB3 5′CL2a favors the bulge configuration observed in the crystal structure since the NMR model was derived for an isolated sD subdomain. We excluded the CVB3 sD NMR model for this comparison as this model included the sequence only after the bulge. Next, we measured the width of the major groove within the P4 helix (Fig. 3e). The measured average distance between the $P_i$ (starting from G47) and $P_{i+6}$ across the P4 helix is 11.5 ± 2.9 Å, which is consistent with the major groove width observed for many other RNA crystal structures (11.1 ± 2.2 Å)[58]. The measured major-groove width for the P2 helix of subdomain B in our crystal structure is also similar to that of the P4 helix (11.6 ± 1.4 Å), suggesting that both the P2 and P4 helices exhibit no significant deviations in the major groove widths, which contradicts the previous claims based on the NMR models of the sD and sB helices that both have a widened major groove to provide interaction sites for the corresponding protein binding. The average major groove width for the previous NMR-based structures is 15.7 ± 4.7 Å[58], perhaps reflecting that the widened major groove widths suggested for the sB and sD helices are due to some ambiguities associated with the NMR modeling.

## Structure of the sD subdomain and 3C^pro binding

Previous biochemical and NMR studies have shown that the sD loop is required and sufficient for the viral 3C^pro binding[14,20,36–38]. Because CVB3 3C^pro binds promiscuously with sD subdomains of other enteroviruses with tetraloops but does not bind with RVB14 sD with a triloop[20], these studies also suggested that 3C^pro recognizes the overall structural feature within the sD loop rather than a defined sequence[20,37,38]. However, none of these previous studies looked into the consequences of highly conserved G66 mutation on 3C^pro binding. Our CVB3 5′CL2a structure solved at 1.9 Å resolution contains a G66C mutation. The construct with the wild-type sD-loop produced crystals,

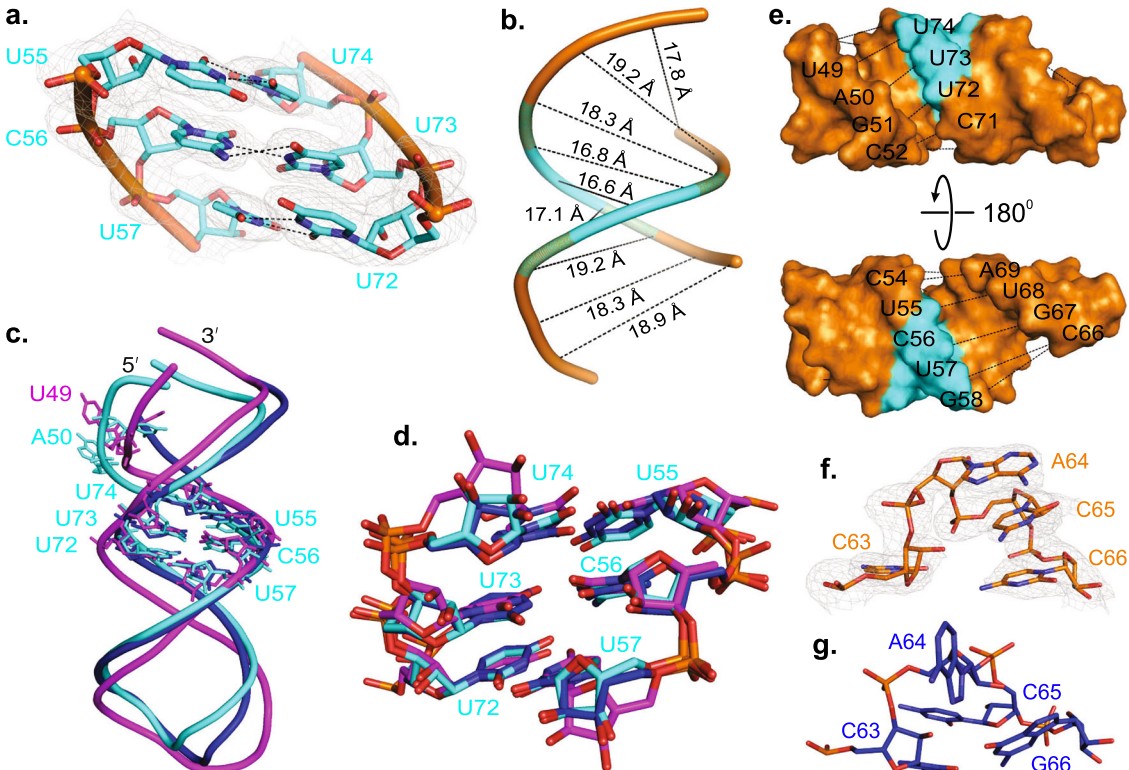

**Fig. 3 | Comparison of the crystal and previous NMR structures of the 5′CL sD subdomain. a** The crystal structure of the Py-Py base-pairs within the P4 helix of CVB3 5′CL2a. **b** The phosphate-to-phosphate distance between the base-pairs across the P4 helix. **c** Superposition of the CVB3 5′CL2a subdomain sD crystal structure (cyan) with previously reported NMR structures of the sD subdomains for CVB3 5′CL (blue) and an enteroviral consensus sequence (magenta). **d** Superposition of only the Py-Py region for the three structures as described in (**c**).

**e** The major groove width within the P4 helix. The dashed black lines depict the distances between the $P_i$ and $P_{i+6}$ across the helix. **f** The GNRA-type tetraloop structure observed in the crystallization 5′CL2a construct. **g** The UNCG-type tetraloop, as observed in the previous NMR structure of the isolated sD subdomain of CVB3 5′CL[38]. Black dashed lines (**a**), (**b**), and (**e**) represents heteroatoms within hydrogen bonding distances. Gray mesh in (**c**) and (**d**) represents the $2|F_o| - |F_c|$ electron density map at contour level 1σ and carve radius 1.8 Å.

but they diffracted to only ~8 Å resolution, suggesting that this mutation was necessary for a robust crystallization. More interestingly, this single mutation appears important for the overall conformation of the sD tetraloop. While the L4 loop, 5′-auCACCgu-3′, within the sD subdomain of our crystal structure adopts a GNRA (N = any nucleotide; R = A or G) type tetraloop fold (Fig. 3f), previous NMR studies with CVB3 sD subdomain and an enteroviral consensus have shown that WT sD loop, 5′-auCACCgu-3′, adopts a UNCG (N = any nucleotide) type tetraloop fold (Fig. 3g)[36,37]. This means the sD loop sequence, especially the fourth nucleotide in the loop, is also crucial to maintain a specific tetraloop conformation of the sD loop, which is consistent with the high conservation of G66 in the enteroviral 5′CLs. Accordingly, the insertion of G as the fourth nucleotide of RVB14 sD triloop to form a tetraloop has been shown to restore its binding to CVB3 3C^pro to a level similar to that of CVB3 5′CL[20].

To understand the structural basis of 3C^pro binding with the 5′CL and to validate the functional relevance of our crystal structure, we tested the binding of a recombinant CVB3 3C^pro with our RNA crystallization constructs. Based on previous studies[61], we mutated the catalytic cysteine (C147) of the 3C^pro to alanine to avoid its potential protease activity. The ITC assays showed that CVB3 3C^pro binds the WT and 5′CL2 constructs with similar affinities (apparent $K_d = 1.40 ± 0.14 μM$ and $1.30 ± 0.09 μM$, respectively), suggesting that grafting of Fab binding sequence in the sB loop had almost no effect on the 3C^pro binding with the sD loop (Fig. 4a, b; see Supplementary Fig. 13 for details). The 5′CL4 construct with the sD loop replaced by the Fab binding pentaloop showed a much weaker affinity (apparent $K_d = 9.4 ± 3.0 μM$ compared to $1.40 ± 0.14 μM$ for the 5′CL2), suggesting that the 3C^pro specifically recognize the sD loop within the CVB3 5′CL

(Fig. 4b; see Supplementary Fig. 13 details). Remarkably, compared to 5′CL2 RNA, the 5′CL2a crystallization construct that contained a G66C mutation within the sD loop showed a much weaker binding with the 3C^pro (apparent $K_d = 17.0 ± 8.0 μM$, Fig. 4b; see Supplementary Fig. 13 for details), suggesting for a critical role of G66 in the 3C^pro protein recognition. However, the relative positioning of G66 in the UNCG-type and C66 in the GNRA-type tetraloop is similar (Supplementary Figure 4), indicating that the tetraloop conformation plays an important role in 3C^pro binding beyond the sequence-specific interactions. Taken together, these analyses support that 3C^pro recognizes a UNCG-type tetraloop within the sD loop, and any mutations that cause alterations to this structure abrogate 3C^pro binding with the enteroviral 5′CLs.

Although the sD loop mutants showed a reduced affinity to 3C^pro, the observable affinities of these mutants suggest additional contacts of 3C^pro with the 5′CL. To test this hypothesis, we performed 3C^pro binding studies with the dinucleotide bulge and Py-Py mutants. The elimination of the dinucleotide bulge by adding the complementary nucleotides (5′CL-sD-NB, Supplementary Fig. 14) showed ~ seven times less affinity (apparent $K_d = 10.1 ± 2 μM$, Fig. 4c) with the 3C^pro compared to the WT 5′CL (apparent $K_d = 1.4 ± 0.14 μM$, Supplementary Fig. 14), indicating direct contacts of the bulge nucleotides with the 3C^pro. However, a construct (5′CL-sD-CN, Supplementary Fig. 14) with Py-Py region replaced by the canonical base pairs displays a similar affinity (apparent $K_d = 3.4 ± 0.7 μM$, Fig. 4c) as the WT 5′CL, supporting that Py-Py region is less likely to be involved in the 3C^pro binding directly. Moreover, consistent with the previous reports[20], an isolated sD subdomain (sDi, Supplementary Fig. 15) binds 3C^pro with a similar affinity (apparent $K_d = 2.1 ± 0.12 μM$, Fig. 4c) as the intact WT 5′CL, suggesting

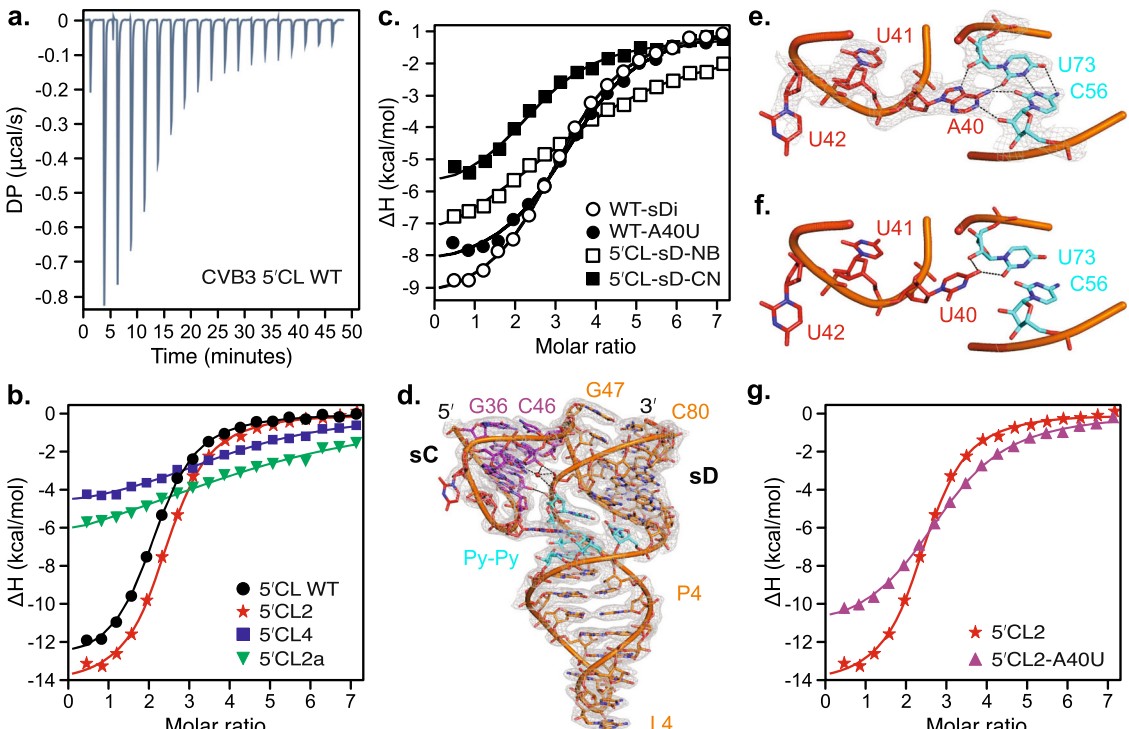

**Fig. 4 | Structure of CVB3 5'CL sD subdomain and its binding interactions with 3 C protein. a** Representative ITC profile for the WT 5'CL. The heat is released upon successive injections of 2 µl of CVB3 3C (~400 µM) to the RNA solution (~10 µM) in the calorimetry cell. **b, c** The binding curves from the ITC data for the binding of the 3C with various crystallization RNA constructs (see Supplementary Fig. 13, 14, and 15 for details). **d** Structure of the CVB3 5'CL2a, showing the tertiary interactions between the sC and sD subdomains, especially the docking of the sC-loop A40 into the Py-Py region of the P4 helix within the sD subdomain. **e** The details of A-minor type tertiary interactions between the A40 and the C56•U73 base-pair within the Py-Py region. **f** An A40U mutation model showing disruption of the A-minor interactions between the sC-loop and the Py-Py helix. **g** The binding curves from the ITC data for the binding of the 3C with the A40U mutant of the parent 5'CL2 construct. Black dashed lines in (**d**), (**e**), and (**f**) represent heteroatoms within hydrogen bonding distances. Gray mesh in d and e represents the $2|F_o| - |F_c|$ electron density map at contour level 1σ and carve radius 1.8 Å.

that the sD subdomain is sufficient for 3C[pro] binding through specific interactions of the sD loop and sD bulge. Furthermore, 3C[pro] binding studies with isolated sD constructs with the canonical base pairing of both U49A50 (sDi-NB, apparent $K_d = 5 \pm 0.1$ µM), U49 only (sDi-U49P, apparent $K_d = 4.9 \pm 0.87$ µM), or A50 only (sDi-A50P, apparent $K_d = 3.5 \pm 0.6$ µM) indicate that the overall structure of the bulge is critical for 3C[pro] binding rather than the identity of the bulged nucleotide (see Supplementary Fig. 15 for the constructs and ITC data). These results are consistent with two different conformations of the bulge being observed in the previous NMR[37] and our crystal structures, and in vitro replication studies showing deletion of both U49A50 but not either U49 or A50 or swapping their positions abrogated the (-)-strand synthesis[14,20].

### Tertiary interactions between the sC and sD subdomains

Our crystal structure of the full-length CVB3 5'CL revealed extensive tertiary interactions between the sC and sD subdomains, including unprecedented interactions between the L3 loop and the P4 helix Py-Py based-paired region. First, the P3 and P4 helix backbones interact through a network of direct and water-mediated hydrogen bonds (Fig. 4d). The G75 and U74 phosphate oxygens make direct hydrogen bonds with 2'OH groups of G44 and C38, respectively. The G75 phosphate oxygens are also involved in water-mediated hydrogen bonding with the G44 N3, 2'OH, and amino groups. Besides these contacts between sC and sD subdomains, interestingly, our crystal structure revealed that the L3 loop A40 docks into the minor groove of the Py-Py helix through the A-minor type interactions (Fig. 4e). The A40 contacts with the C56 and U73 through Watson-Crick–Sugar-Edge and Hoogsteen–Sugar-Edge hydrogen bonding interactions,

forming an A40•C56•U73 base-triple. Specifically, the C56 and U73 2' OH groups interact with the A40 N1 and N7, respectively, and their O2s with the A40 amino group through direct hydrogen bonds (Fig. 4e). To validate this docking of the A40 in solution, we conducted NMR studies for the 5'CL2 construct. By applying different variations of $A^{2R}U^{6R}$ labeling schemes (see Methods), we observed NOE (Nuclear Overhauser Effect) signals between the A40 H2 and H1's of two Uracils (Supplementary Fig. 16), which is consistent with the A40 and Py-Py interactions in our crystal structure, where the H1' of U57 and U74 are close to H2 of A40 (<4 Å, Supplementary Fig. 16).

The divalent cations such as Mg[2+] are known to stabilize the RNA tertiary structures. Nevertheless, the presence or absence of Mg[2+] in the solution did not alter the binding affinities of 3C[pro] with the 5'CL constructs significantly (for the 5'CL2, apparent $K_d$ with 10 mM Mg[2+] = $1.4 \pm 0.2$ µM compared to $1.30 \pm 0.09$ µM in an Mg[2+] dialyzed solution), supporting that the long-range tertiary structures, compared to the secondary structural features, within the 5'CL may have no major effect on 3C[pro] binding. Notably, the 1D and 2D NMR spectra for the CVB3 5'CL2a construct in a solution with 5 mM Mg[2+] and an Mg[2+] dialyzed solution are almost superimposable (Supplementary Fig. 17). Only small salt-dependent chemical shifts, but no major shifts or changes in the intensity of peaks due to larger structural changes, are observed, suggesting that, unlike the RVB14 5'CL[20], the CVB3 5'CL folds into the compact H-type 4-way junction independent of the Mg[2+] in the solution. Consistent with the NMR results, we did not observe any prominent Mg[2+] binding sites in the crystal structure even at 1.9 Å resolution, although the crystallization condition contained at least 10 mM Mg[2+].

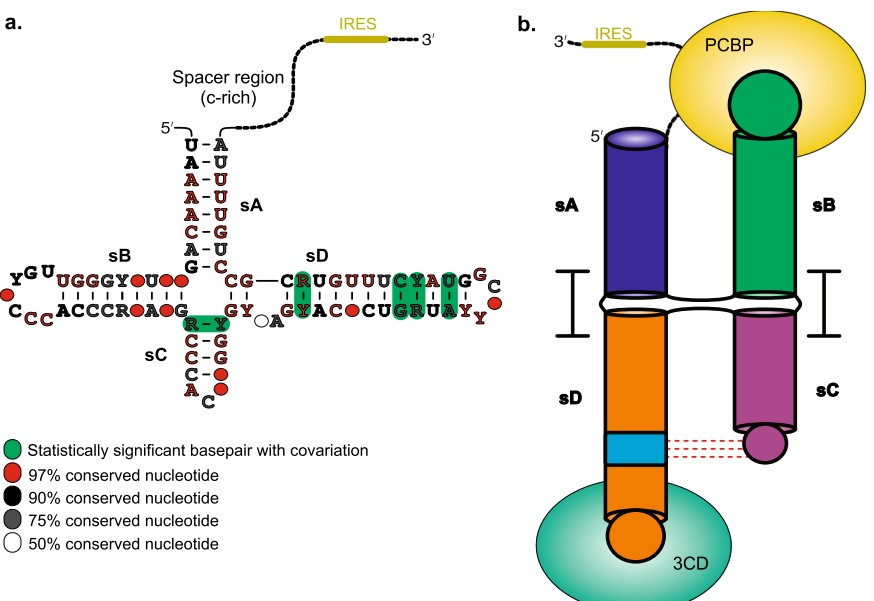

**Fig. 5 | Consensus structural model of the enteroviral 5′CLs. a** Proposed secondary structural model of consensus enteroviral 5′CL (also see Supplementary Fig. 20 for details). The 5′CLs are one of the most conserved RNA elements among enteroviruses (**b**) The three-dimensional consensus fold of the 5′CLs in enteroviral genomes based on the crystal structure of CVB3 5′CL showing binding modes of viral 3CD and host PCBP proteins. The red dashed lines represent tertiary interactions and depict coaxial stacking of the corresponding subdomains.

## Potential roles of A40·Py-Py interactions in 3C$^{pro}$ and PCBP2 proteins binding

The long-range interactions between highly conserved A40 and Py-Py nucleotides among enteroviral 5′CLs suggest that these nucleotides may have roles important for 3C$^{pro}$ binding with the 5′CL structure. Predictably, the A40 or Py-Py mutation would destabilize the 5′CL tertiary structure (see Fig. 4f) and reduce the binding affinity of 3C$^{pro}$. However, the A40U mutation within the sC-triloop of CVB3 5′CL2 had little effect on the 5′CL binding with 3C$^{pro}$ (apparent $K_d = 3.10 \pm 0.22\,\mu$M compared to $K_d = 1.40 \pm 0.14\,\mu$M for the 5′CL2 (Fig. 4g; Supplementary Fig. 14). Yet, this slight decrease in the affinity for the A40U mutant may be due to minor conformational changes within the 5′CL structure induced by this A40U mutation (see Fig. 4f). These results also agree with our ITC measurements for intact wild-type 5′CL and an isolated sD subdomain for 3C$^{pro}$ binding (apparent $K_d = 1.4 \pm 0.14\,\mu$M and $2.10 \pm 0.12\,\mu$M, respectively; Fig. 4b, c, Supplementary Figs. 13 and 15), as well as previously reported affinities of these RNA constructs for 3C$^{pro}$ based on a filter-binding assay (apparent $K_d = 2.7$ and $4.6\,\mu$M, respectively)[20], supporting that the A40 interactions have no direct role in binding the 3C$^{pro}$. Similarly, the 3C$^{pro}$ binds the wild-type 5′CL, and the 5′CL-sD-CN (the Py-Py replaced by canonical pairs) constructs similarly (Fig. 4c; Supplementary Figs. 14 and 15), consistent with the previous observation that the replacement of the Py-Py helix with canonical base-pairs did not abolish the (-)-strand synthesis[14]. However, as shown in a yeast three-hybrid assay, the deletion of U74 abrogated, but the compensatory C56U mutation still preserved the 3C$^{pro}$ binding with the 5′CL[20]. Taking together our structural and 3C$^{pro}$ binding results, it is apparent that the integrity of the dinucleotide bulge and the loop that preserve the overall structure of the sD subdomain is required for effective 3C$^{pro}$ binding, and the A40U·Py-Py tertiary interactions have consequences beyond the 3C$^{pro}$ binding.

We hypothesized that a more prominent structural role of these interactions is to stabilize the 4-way junction of the 5′CL that positions the 3C$^{pro}$ binding site (sD loop) away from the host PCBP binding site (sB loop) to avoid potential steric clashes between 3C$^{pro}$ and PCBP during the replication machinery assembly. To test this hypothesis, we recombinantly expressed and purified the full-length human PCBP2 protein and performed the binding assays using gel electrophoresis (Supplementary Fig. 18). As PCBP2 has been shown to bind both the sB loop and the C-rich spacer region between the 5′CL and IRES, we also performed binding tests with the longer RNA constructs that contain both the 5′CL and spacer region. Consistent with the previous studies that PCBP2 recognizes the sB loop, we observed that the WT 5′CL but not the sB loop mutant 5′CL2 bind PCBP2. Interestingly, the A40U mutant, which had no effect on 3C$^{pro}$ binding, interacts with the PCBP2 with much less affinity compared to the WT 5′CL, suggesting that the A40·Py-Py interaction has some roles in PCBP2 binding to the 5′CL structure. Strikingly, the longer RNA construct 5′CLWTSP with the sB loop and spacer sequence binds the PCBP2 tightly compared to the 5′CL without the spacer, indicating that the sB loop and the spacer sequence reside close to each other, which is consistent with the observed juxtaposition of sA and sB subdomains in our crystal structure. Although the effect of A40·Py-Py interaction on the PCBP2 binding and 5′CL function remains to be studied thoroughly, our preliminary results suggest that the H-shaped 5′CL structure stabilized this A40·Py-Py long-range contact juxtapose the PCBP2 binding sites, the sB loop, and the spacer well-separated from the 3C$^{pro}$ binding site, the sD subdomain.

## Bioinformatics analysis of the enteroviral 5′CL sequences

To study the structural relevance of the nucleotides within enteroviral 5′CLs, we performed phylogenetic analysis using bioinformatics tools. The alignment of over 5000 enteroviral genome sequences shows a high degree of sequence conservation within the 5′CLs region (Supplementary Fig. 19). The computed consensus secondary structure (Fig. 5a; Supplementary Fig. 20) from these sequence alignments for seven enteroviral genotypes using R-Scape[62] indicates that the structural features within each subdomain sA, sB, sC, and sD are highly conserved, consistent with the previously proposed consensus secondary structure of the enteroviral 5′CL in the Rfam database[63] obtained by aligning 162 sequences from 87 enterovirus species only. Notably, the key nucleotides observed in our CVB3 5′CL crystal structure, including the A40, Py-Py, and sD bulge, are absolutely conserved, suggesting the viral requirement to maintain these nucleotides against

strong selection pressure. Interestingly, the R-scape[62] optimized consensus secondary structure predicts the sC subdomain as a four-base-pair helix closed by a triloop with the absolutely conserved A40 as the third nucleotide within the triloop. Similarly, the sD subdomain contains a highly conserved 6 base-pair helix between the four-way junction and Py-Py base-paired region. The helix is also interrupted by a two-nt bulge between the 2nd and 3rd base-pairs following the four-way junction. The length and secondary structure conservation within the sC and sD subdomains (Supplementary Fig. 21) with this specific arrangement is perhaps required to maintain the specific interactions between the L3 loop and the Py-Py based-paired region, consistent with the tertiary interactions observed in our crystal structure.

## Discussion

There are two distinct steps in enteroviral genome replication. The first is the synthesis of (-)-strand RNA using the genomic RNA as a template, and the second is the back synthesis of the (+)-strand genomic RNA using the newly made (-)-strand RNA as a template. The 5′CL is one of the major components of the enteroviral genome replication machinery that directly binds viral and cellular protein factors to form a replication-competent RNP complex. The sD subdomain of 5′CL interacts with viral 3CD fusion protein through its 3C$^{pro}$ that brings the viral polymerase D$^{pol}$ to the replication initiation site. The sB subdomain binds PCBP2, which then interacts with PABP-poly(A) tail complex at the 3′end to circularize the viral genome. The 3CD also interacts with cre, a stem-loop RNA domain located within the 2C coding region, to promote the uridylation of viral VPg protein. The resulting VPg-pUpU then serves as a primer for the (-)-strand RNA synthesis by active D$^{pol}$ polymerase, an autocleavage product of the 3CD precursor. Additionally, the conserved RNA sequence and structural features in the 5′CL influence the VPg uridylation and viral genome stability and may facilitate the replication machinery to recognize the enteroviral genome in a milieu of myriad cellular RNAs[64]. Therefore, the high-resolution structures of the 5′CLs and the corresponding RNP complexes with 3C or 3CD and PCPB provide important information not only for a detailed understanding of mechanisms of enteroviral genome replication–a poorly understood virological process–but also for developing drugs that target this platform. Although previous virological, molecular biology, biochemical and biophysical approaches have facilitated some structural and functional understanding of the 5′CLs for initiating enteroviral genome replication and how their subdomains interact with the 3C or 3CD and PCPB proteins at the molecular levels[36–44], our CVB3 5′CL crystal structure represents the first high-resolution structure of an intact 5′CL from any enterovirus, setting up a groundwork for further investigation on the 5′CL-mediated enteroviral replication mechanisms.

The overall secondary structures of the CVB3 and RVB14 5′CLs, including the NMR-derived models for the isolated sC and sD subdomains, are almost identical. However, the 3-dimensional architecture of our CVB3 5′CL crystal structure differ significantly compared to the RVB14 5′CL models. Unlike the RVB14 5′CL model[43], our crystal structure adopts an H-type 4-way junction with the coaxially stacked sA-sD and sB-sD subdomains, juxtaposing the sA with sB and sC with sD. These discrepancies are perhaps related to the low resolution of the SAXS technique and computational modeling of the intact 5′CL structure based on the NMR structures of isolated sB and sD subdomains only without considering the structures of the sA and sC subdomains. Moreover, using the NMR and SAXS approaches, Warden et al.[43] previously proposed the Mg$^{2+}$-dependent structural models for the full-length RVB14 5′CL. Without the Mg$^{2+}$, the RVB14 5′CL assumed an extended conformation similar to a cK-type 4-way junction with the sB and sD subdomains residing almost perpendicular to one another, but with the Mg$^{2+}$, the same RNA folded in a more compact conformation that juxtaposed the sB and sD subdomains. Moreover, RVB14 5′CL displayed a substantial change in the NMR chemical shifts

for the nucleotides involved in the Py-Py helix upon the addition of the Mg$^{2+}$[43], which was not consistent with that observed for the same sD subdomain in isolation[40]. However, these observations are consistent with the stabilization of the Py-Py region through tertiary interactions with the loop L2 as observed in our intact CVB3 5′CL crystal structure. Although Mg$^{2+}$ dependent conformations of various enteroviral 5′CLs are yet to be studied in detail, unlike previous results with RVB14, our preliminary NMR results support that the CVB3 5′CL folds into compact H-type 4-way junction structure independent of the Mg$^{2+}$ in the solution, suggesting the different roles of Mg$^{2+}$ in stabilizing different enteroviral 5′CLs.

Previous studies have shown that the efficacy of enteroviral genome replication depends on the sites of interaction for PCBP and 3C or 3CD proteins within the 5′CL structure. The sD subdomain is required and sufficient for the 3C binding. Consequently, the bulge, Py-Py region, and tetraloop structures are the most conserved regions within the sD subdomain that significantly influence both (+)- and (-)-strand RNA synthesis. For CVB3, the deletion of the dinucleotide bulge U49A50 inhibited the (-)-strand synthesis, but mutations A49U50 to swap these nucleotide positions and deletion of U49 or A50 nucleotide had almost no effect, suggesting that a single nucleotide bulge is sufficient for the (-)-strand synthesis. However, the (+)-strand synthesis required an intact dinucleotide bulge. While deletion of the U74 within the Py-Py helix abolished the 3C binding with the 5′CL, previous NMR data did not support direct interactions of the protein with the nucleotides in the Py-Py region. However, replacing the C56•U73 with the U56•U73 pair preserved the 3C binding with the 5′CL. Further studies using cell-free assays for PV and CVB3 showed that complete elimination of Py-Py mismatch with Watson-Crick base-pairs increased (-)-strand synthesis while decreasing the (+)-strand synthesis, suggesting that the Py-Py region in the sD subdomain is essential to maintain the ratio of (-) and (+) RNA strands during viral infection. Structurally, the Py-Py region maintains an A-form helicity, any mutations that do not alter the A-form helicity would still permit the A-minor type interactions between the sC loop and Py-Py region, but a similar scenario is less likely in an analogous structure in the 3′-end of the (-) strand. Perhaps the requirement for the Py-Py region in the 5′CL is more prominent in the (+)-strand synthesis using a (-)-strand RNA template. It is beyond the scope of this study to study the structures of the analogous RNA structures formed within the 3′-end of the (-) strand RNA, but our crystal structure supports that the Py-Py region stabilizes the overall 5′CL structure through tertiary interactions with the sC loop. Also, the isolated sD subdomain shows similar binding affinity as the intact CVB3 5′CL, suggesting that the absolutely conserved structural features in the sD subdomain among enteroviruses are important beyond the binding of the 3C protein. Moreover, although the effects of sC loop mutations, including the A40, in (-)- and (+)-strand synthesis are yet to be investigated, absolute conservation of the A40 against intense selection pressure underscores its requirement to maintain the integrity of these tertiary interactions for the viral genome replication.

A comparison of our CVB3 5′CL structure with other enteroviruses and rhinoviruses suggests that lengths of the P3 helix (four base-pairs) and the P4 helix preceding the Py-Py region (six base-pairs interrupted by the dinucleotide bulge) are the most conserved structural features. These features underscore a specific requirement for positioning the A40 and Py-Py pairs to facilitate tertiary interactions between the sC and sD subdomains. These tertiary interactions within the 5′CL are more pronounced when considering the PCBP binding sites. Within the enteroviral 5′CL, PCBP interacts with the sB loop and a C-rich spacer sequence at the end of the 5′CL structure, the later sequence contributing more to the binding interactions. The mutations to C-rich sequences within the subdomain B and the spacer region abolish PCBP binding and deteriorate the (-)-strand RNA synthesis, supporting that PCBP binding in these regions is critical for efficient replication of the enteroviral genome[19,27,28]. Unlike previously proposed models of RVB14

5′CL with the juxtaposition of sD and sB subdomains, our crystal structure of CVB3 5′CL revealed the juxtaposition of sA and sB subdomains, bringing the sB loop and the C-rich spacer sequence close to each other (Fig. 5b), which is consistent with our preliminary binding studies with human PCBP2 and supports those previous biochemical observations. Therefore, the tertiary interactions between the sC loop and the Py-Py regions play important roles in stabilizing the 5′CL structure and precisely positioning the sB and sD loops, providing a ready-made platform for replication components to bind rather than in direct involvement of these substructures with the 5′CL-protein interactions. However, further studies to determine the high-resolution structures of enteroviral 5′CLs in complex with corresponding viral protein 3C or 3CD and host protein PCPB2 will set critical steps towards a mechanistic understanding of the enteroviral replication and developing potential therapeutics targeting this enteroviral replication platform.

## Methods

### RNA synthesis and purification

RNA constructs for crystallization, protein binding, and NMR studies were synthesized by in vitro transcription. DNA template with a T7 promoter sequence for transcription reaction was produced by PCR amplification of ssDNA purchased from Integrated DNA Technologies. The first two nucleotides of reverse primer were 2′-OMe modified to reduce the 3′-end heterogeneity of the transcript[65]. RNA was transcribed for 3 h at 37 °C in a buffer containing 40 mM Tris-HCl, pH 8.0, 2 mM Spermidine, 10 mM NaCl, 25 mM $MgCl_2$, 1 mM DTT, 40 U/ml RNase inhibitor, 5 U/ml TIPPase, 5 mM of each NTP, 50 pmol/ml DNA template, and 50 µg/ml homemade T7 RNA polymerase[66]. Transcription reaction was quenched by adding 10 U/ml DNase I (Promega) and incubating at 37 °C for 1 h. For synthesizing NMR samples, RNA was transcribed using the labeled NTPs purchased from Cambridge Isotope Laboratories. RNA was transcribed for 8 h, and the reaction was quenched with 500 mM Urea and 60 mM EDTA, followed by boiling the mixture for 5 min. All RNA samples were purified by denaturing polyacrylamide gel electrophoresis (dPAGE). The band was visualized by UV shadowing, excised from the gel, and eluted overnight at 4 °C in 10 mM Tris, pH 8.0, 2 mM EDTA, and 300 mM NaCl. The eluted RNA's buffer was exchanged with pure water three times using a 10 kDa cut-off Amicon column (Millipore Sigma). RNA was collected, aliquoted into 300 µl fractions, and stored at −80 °C until further use. For the NMR samples, RNA was further purified by ethanol precipitation before getting lyophilized and buffer exchanged for the NMR measurements.

### Fab expression and purification

Fab BL3-6 expression plasmid was a kind gift from Joseph Piccirilli, the University of Chicago. The Fab was purified according to the published protocols[52,67,68]. Briefly, the plasmid was transformed into 55244 E. coli competent cells and streaked into an LB-agar plate with 100 µg/ml of carbenicillin. Several colonies were selected to inoculate a 15 ml starter culture and grown at 30 °C for 8 h. The starter culture was then used to inoculate 1 liter of 2xYT media, and cells were grown for 24 h at 30 °C. For Fab overexpression, the cells were centrifuged at 22 °C and 6000 × g for 10 min, resuspended in 1-liter phosphate-depleted media, and grown for 24 h at 30 °C. The cells were harvested by centrifugation at 4 °C and 6000 × g for 10 min, resuspended in PBS, pH 7.4 buffer with 0.01 mg/ml bovine pancreas DNase I (Sigma-Aldrich), and 400 mM Phenylmethylsulfonyl fluoride (PMSF), and lysed by sonication (Qsonica, Cole-Parmer). The mixture was first centrifuged at 40,000 × g, the clear lysate was filtered through a 0.45-micron filter (VWR), and the Fab was purified using the Bio-Rad NGC fast protein liquid chromatography (FPLC) system. First, the lysate was passed through a Hi-trap protein A column (Cytiva), and the captured Fab was eluted with 0.1 M acetic acid. The fractions were collected, diluted 10x using PBS, pH 7.4 buffer, and loaded into a Hi-trap protein G column (Cytiva). The eluted Fab fractions from the protein G column in 0.1 M glycine, pH 2.7, were collected, diluted 10x with a 50 mM NaOAc, 50 mM NaCl, pH 5.5 buffer, and loaded into a Hi-trap heparin column (Cytiva). Finally, the Fab fractions eluted from the heparin column by the gradient elution using 50 mM NaOAc, 2 M NaCl, pH 5.5 buffer were collected, and buffer exchanged 3x with 1x PBS pH 7.4 using 30 kDa cut-off Amicon column (Millipore Sigma). The concentrated Fab was collected, analyzed by 12% SDS-PAGE, and tested for RNase activity using the RNaseAlert kit (Ambion, www.thermofisher.com). Aliquots (-300 µl) of purified Fab were stored at −80 °C.

### Native gel electrophoresis

RNA in water (-100 ng) was refolded in a buffer containing 10 mM Tris-HCl, pH 7.4, 10 mM $MgCl_2$, and 100 mM NaCl. RNA was heated at 90 °C for 1 min, and an appropriate volume of 10x refolding buffer was added, followed by incubation at 50 °C for 10 min and then in ice for 5 min. The refolded RNA was then incubated for 30 min at room temperature with different equivalents of Fab or 3 C protein. The protein-RNA complex samples were mixed with an appropriate volume of 6x native agarose gel loading solution containing 30% glycerol, 0.1% each bromophenol blue, and xylene cyanol. These samples were loaded onto 10% native polyacrylamide gels and run at 115 V in pre-cooled 0.5x TBE buffer (50 mM Tris-base, 50 mM boric acid, and 1 mM EDTA, pH 7.5) at 4 °C. The gels were stained with ethidium bromide and imaged using the Azure 200 gel documentation system (Azure Biosystems).

### Crystallization

The RNA sample was refolded in a folding buffer containing 10 mM Tris-HCl, pH 7.4, 10 mM $MgCl_2$, and 100 mM NaCl. First, RNA in water was heated at 90 °C for 1 min, and an appropriate volume of 10x refolding buffer was added, followed by incubation at 50 °C for 10 min and in ice for 5 min. The refolded RNA was then incubated for 30 min at RT with 1.1 equivalent of the Fab and concentrated to 6 mg mL$^{-1}$ using a 10 kDa cut-off, Amicon Ultra-1 column (Millipore Sigma). Then, Fab −RNA complexes were passed through 0.2 µm cut-off Millipore centrifugal filter units (www.emdmillipore.com). The Xtal3 Mosquito liquid handling robot (TTP Labtech, ttplabtech.com) was used to set up hanging-drop vapor-diffusion crystallization screens at RT using commercially available screening kits from Hampton Research and Jena Bioscience. The crystals were formed within a week in various conditions. Select conditions were further optimized for pH, precipitant, and salt concentration to grow larger crystals using the hanging drop vapor diffusion method. The crystals grew to full size over a week. For cryoprotection, drops containing suitable crystals were brought to 30% glycerol without changing the other compositions. Crystals were immediately flash-frozen in liquid nitrogen after being fished from the drops and taken to Argonne National Laboratory for collecting the X-ray diffraction data.

### Structural data collection, processing, and analysis

The X-ray diffraction data sets were collected at the Advanced Photon Source NE-CAT section beamlines 24-ID-C and 24-ID-E. All the datasets were then integrated and scaled using its on-site RAPD automated programs (https://rapd.nec.aps.anl.gov/rapd/). Initial phases were obtained by molecular replacement with the previously reported structure of Fab BL3-6 (PDB: 6B14 as the search model using Phaser on Phenix[69]. Iterative model building and refinement were performed using COOT[70], and the Phenix package[69]. RNA was built unambiguously by modeling the individual nucleotides into the electron density map obtained from the molecular replacement. During the refinement, default NCS options and auto-selected TLS parameters in Phenix were used. Most water molecules were automatically determined by Phenix software during refinements. However, some water molecules were

added manually for the positive electron density in the map based on their possibility of forming hydrogen bonds with protein or RNA residues. Solvent-accessible surface area and area of interaction were calculated using (http://www.ebi.ac.uk/pdbe/pisa/)[57]. Structure-related figures were made in PyMOL (The PyMOL Molecular Graphics System, Version 2.0 Schrödinger, LLC), and figure labels were edited in Corel-Draw (Corel Corporation, http://www.corel.com).

### CVB3 3C$^{pro}$ expression and purification

The recombinant CVB3 3C$^{pro}$ was expressed and purified using previously described protocols with some modifications[20,38,61]. The codon-optimized DNA sequence encoding for the CVB3 3C$^{pro}$ C147A mutant was cloned into pET-22b(+) vector between NdeI and XhoI restriction sites (GenScript, https://www.genscript.com). The expressed protein contained the sequence MGPAFEFAVA MMKRNSSTVK TEYGEFTMLG IYDRWAVLPR HAKPGPTILM NDQEVGVLDA KELVDKDGTN LELTLLKLNR NEKFRDIRGF LAKEE-VEVNE AVLAINTSKF PNMYIPVGQV TEYGFLNLGG TPTKRMLMYN FPTRAGQAGG VLMSTGKVLG IHVGGNGHQG FSAALLKHYF NDEQ with the vector encoded 6x-His tag at the C-terminal. The expression plasmid was transformed into BL21 (DE3) E.coli, and the cells were cultured in a 2xYT medium supplemented with 100 µg/mL Carbenicillin at 37 °C with 220 rpm shaking until the OD of ~0.6. The cells were then induced for protein expression by using IPTG (isopropylthio-β-galactoside) to the final concentration of 0.5 mM, grown for 6 hours at 25 °C, and finally, harvested by centrifugation at 6000 x g. The protein purification was carried out in a Bio-Rad FPLC system. The cell pellet was resuspended in a lysis buffer containing 50 mM Tris HCl, pH 7.5, 300 mM NaCl, and 5 mM imidazole and lysed using sonication. The lysate was centrifuged at $40,000 \times g$ at 4 °C, and the supernatant was passed through a 0.45-micron filter. The clarified lysate was then applied to a HisTrap™ column (Cytiva), and the protein was eluted from the column with a buffer (50 mM Tris HCl, 300 mM NaCl, and 250 mM imidazole, pH 7.5) after washing the column with 5 column volumes of the lysis buffer. The eluted fractions were collected, dialyzed against a buffer containing 50 mM Tris HCl, 100 mM KCl, 1 mM EDTA, and 5% Glycerol, pH 7.5, and purified further by size-exclusion chromatography (HiLoad® 26/600 Superdex® 200 pg column, Cytiva). The single-peak protein fractions were pooled and concentrated using the Amicon centrifugal filters (molecular weight cut-off 10 kDa, Millipore Sigma), flash froze with liquid N$_2$ in small aliquots, and stored at −80 °C.

### Human PCBP2 expression and purification

The full-length human PCBP2 (residues 11–359) with C-terminal 6x-His tag was expressed and purified using previously described protocols with some modifications[28,71]. Briefly, the codon-optimized DNA sequence encoding the human PCBP2 protein was cloned into the pET-22b(+) vector between NdeI and XhoI restriction sites (GenScript, https://www.genscript.com). The expression and purification were performed using a similar procedure as discussed above for 3C$^{pro}$. The lysis buffer contained 50 mM Tris (pH 7.5), 300 mM NaCl, and 20 mM imidazole, whereas the HisTrap column elution buffer contained 50 mM Tris (pH 7.5), 300 mM NaCl, and 250 mM imidazole. The eluted fractions were collected, dialyzed against a buffer containing 50 mM Tris (pH 7.5), 100 mM KCl, 1 mM EDTA, and 5% glycerol, and purified further by size-exclusion chromatography with HiLoad® 26/600 Superdex® 75 pg column. The single-peak protein fractions were pooled and concentrated using the Amicon centrifugal filters (molecular weight cut-off 30 kDa), flash froze with liquid N$_2$ in small aliquots, and stored at −80 °C until further use.

### Isothermal titration calorimetry

The isothermal titration calorimetry (ITC) experiments were carried out in the MicroCal PEAQ-ITC Automated (Malvern Panalytical) equipment using freshly prepared RNA and protein samples. The RNA and protein samples were dialyzed overnight into a buffer containing 50 mM Tris HCl, pH 7.5, 100 mM KCl, 1 mM EDTA, and 5% Glycerol. The injection syringe contained 150 µl of ~400 µM 3C$^{pro}$ protein, and the calorimetry cell was loaded with 500 µl of ~10 µM RNA. After thermal equilibration at 25 °C and an initial 60-second delay, a single injection of 200 nl followed by 19 serial injections of 2 µl 3C$^{pro}$ protein was made into the calorimetry cell. The reported $K_d$ for each RNA construct represents the average ± standard deviation obtained from the triplicate experiments.

### NMR data acquisition, processing, and analysis

The RNA samples were measured in 100% D$_2$O (99.8%, Cambridge Isotope Laboratories) with 20 mM sodium phosphate, pH 7.4, 70 mM KCl, 5 mM NaCl, and 5 mM MgCl$_2$ (unless described differently) in a 5 ml tube at concentrations varying from 0.7 to 1.2 mM. The data was collected with a Bruker AVANCE spectrometer (600 MHz, $^1$H) at 308 K with a weighted sampling scheme[72]. Non-exchangeable $^1$H assignments were obtained from 2D NOESY data (NOE mixing time = 300 ms, relaxation delay = 12.0 s, T = 308 K). All NMR data were processed with NMRFX[73] and analyzed with NMRViewJ[74]. Partial assignments were made using the standard NOE-based sequential assignment strategy[75,76] on A$^{2R}$U$^R$, U$^{6R}$, and A$^{2R}$ labeled RNA samples following the strategies that were pioneered in earlier studies[77]. The assignments were validated by comparisons with chemical shift values in the Bio-MagResBank NMR repository using NMRViewJ[78–80]. Additionally, previous assignments were used to transfer and cross-validate some of our assignments[36,37].

### Statistics and reproducibility

Unless specified otherwise, experiments associated with native PAGE, SDS PAGE, and ITC were performed in triplicate. Each replicate produced similar results.

### Reporting summary

Further information on research design is available in the Nature Portfolio Reporting Summary linked to this article.

## Data availability

Atomic coordinates and structure factors for the reported crystal structure have been deposited with the Protein Data Bank under accession code 8DP3. The authors will provide the raw data, additional information, and materials, including the plasmid for Fab BL3-6 expression, upon request. The requests should be addressed to D.K.

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

## Acknowledgements
This work was supported by start-up, SURFF, and START awards from the University of Maryland Baltimore County and the NSF CAREER award 2236996 to D.K. and partly by the NIH grant T32 GM066706 to N.K.D. The crystallographic work is based on research conducted at the Northeastern Collaborative Access Team beamlines (24-ID-C and 24-ID-E), funded by the National Institute of General Medical Sciences from the National Institutes of Health (P30 GM124165). The Eiger 16 M detector on 24-ID-E is funded by an NIH-ORIP HEI grant (S10OD021527). This research used the Advanced Photon Source resources – a U.S. Department of Energy (DOE) Office of Science User Facility operated for the DOE Office of Science by Argonne National Laboratory under Contract No. DE-AC02-06CH11357. The authors would like to thank the staff of the Advanced Photon Source at Argonne National Laboratory for providing technical advice during data collection. We are thankful to Prof. Michael Summers, the University of Maryland Baltimore County, for kindly providing the NMR and ITC facilities and for the constructive comments on the manuscript.

## Author contributions
N.K.D. and D.K. conceived and designed the experiments. N.K.D. prepared the samples, conducted most of the experiments, and solved the crystal structures with the help of D.K. N.M.H. collected and analyzed the NMR data. J.V. performed all bioinformatics and computational work. S.S. designed and set up initial crystallization trials. H.A.B. and M.O. helped N.K.D. with biochemical experiments and collecting X-ray diffraction data. N.K.D. analyzed most of the biochemical and crystallographic data and interpreted the results with D.K. N.K.D. and D.K. wrote the manuscript, and all authors critically reviewed the manuscript.

## Competing interests
The authors declare no competing interests.
