## [Peer Review File · Nature Communications]

Crystal structure of a highly conserved enteroviral 5' cloverleaf RNA replication elementREVIEWER COMMENTS

Reviewer #1 (Remarks to the Author):

The authors present a crystal structure and additional analysis of a (mutated) 5' RNA clover leaf from the start of the Cocksackievirus B3 genome, in complex with a Fab fragment. This cloverleaf is central in the genomic replication of the virus.

The three most interesting results are (i) the overall architecture of the clover leaf, with SA and SD coaxial, and SB and SC coaxial, and these two long helices parallel to each other and in close contact, forming a compact structure.

This architecture has some similarity but some contrast with a previous NMR/SAXS-based study of the similar region from RV14. The similarity is that SB and SD are parallel but not coaxial to each other in both studies. However, in the NMR/SAXS study, SB and SD are in close contact, while in the present study, SB and SD are at opposite ends of the structure, and in fact, are anti-parallel to each other rather than parallel. The authors suggest that this finding may be related to the 2nd very interesting result, (ii) the observation of interactions between the pyrimidine-pyrimidine base pair region in the center of SD, and the loop region at the end of SC. The presence of SD-SC interactions, and therefore the overall clover leaf architecture, is further supported by NMR analysis of NOEs. The third interesting point is that (iii) this compact coaxial structure was observed even in the absence of magnesium, which contrasts with the marked magnesium-dependence seen in the RV14 study.

This is an interesting study worthy of publication, once the following items are addressed:

Main comments

While some differences between the (present) Cocksackievirus structure and the previous rhinovirus structure can (and are) ascribed to differences in techniques of X-ray crystallography vs. NMR/SAXS, and some to the different sequences of the RNA molecules (which was also discussed by the authors), the following additional factors should also be discussed:

(i) The current study utilizes a mutant RNA molecule (5'CL2 G66C) in which the loop at the end of SB has been mutated to be complimentary to a Fab fragment required for co-crystallization, and the loop on the end of SD was also mutated to improve crystallization results.

(ii) The presence of the Fab fragment in the co-crystal

(iii) Crystal contacts in general

(iv) In particular, crystal contact between A50, near the four-helix junction region, and a symmetry-related Fab fragment

The major change needed is an earnest discussion of the role that the above factors may play in affecting what may be delicately balanced conformational stabilities. This discussion is necessary in order to properly assess the differences between the current CVB3 structure and the previous RV14 5'CL structure. For instance, interactions involving loop C are proposed to be critical for the observed conformation, but loop B and loop D have been mutated, so there is potential for change of interactions. How can it be known that interactions with loop B and loop D are not important? A related minor point: discussion of how the G66C mutation was identified as assisting crystallization would be helpful.

Other comments/questions:

Line #

213-228

discussion of widened groove in rhinovirus SB region should focus on the smaller number of base pairs in rhinovirus SB. This is mentioned later on in the paragraph, but it should be the main point, and thus the CVB3 SLB result does not present a substantial difference from the RV14 SLB result. Or at least, the difference is due to different number of base pairs, rather than different techniques, etc

362-364

The authors suggest that the minimal effect of magnesium on binding to 3C protease provides evidence that the CVB3 conformational architecture has no dependency on magnesium. I do not see that this follows, since all evidence points to 3C protease interacting only with SD and its loop, an interaction which would not manifestly be dependent on tertiary structure (relative alignment of helices). Of course the authors provide other evidence that CVB3 does not seem to show this magnesium dependence, but 3C binding is not a convincing part of this evidence.

368

Reference is made to 2D NMR data in Supp. Fig. 10, but this Fig. is only a small region of a 1D NMR spectrum. This argument would be far more convincing with the 2D data of the imino proton region, either a NOESY or an 15N-HSQC.

401-407

If the purine-purine base pairs are needed to position SD along-side SC, and thus to position SD and SB in an anti-parallel fashion, then why are the purine-purine base pairs dispensable for (-)-strand synthesis, which uses this structure as a platform?

Typos

Line #

60 consensus

264 check Figure reference is correct

Reviewer #2 (Remarks to the Author):

Summary:

Das et al present a crystal structure of the 5' cloverleaf sequence from coxsackievirus B3, an important enterovirus which also serves as a model system for several other medically important viruses including poliovirus and hepatitis A. The 5' cloverleaf sequence is an essential RNA element within enteroviruses necessary for viral replication and pathogenicity.

The authors used a strategy of including a Fab binding region into their RNA construct to favor the formation of crystal contacts. Importantly, they demonstrate that the location of their Fab binding site does not disrupt the enterovirus protein binding activity of the RNA. Another single point mutation was made which increased resolution. This point mutation did affect protein binding, however an analysis of the crystal structure would suggest that it is unlikely to cause a global change in RNA structure that would affect the overall conclusions of the paper.

The resolution of the solved structure is high (1.9Å) permitting unambiguous assignment of electron density. The authors report only a single RNA-RNA crystal contact suggesting that the observed structure should lack crystal artifacts, however the paper should include a discussion of protein-RNA crystal contacts in order to confirm that they are not perturbing the global RNA structure. Supplementary figure 5 is a useful starting point, however there should be an inset to show all other significant crystal

contacts within the RNA structure. It would also be useful to include a color-map of b-factors somewhere in the supplement.

The authors present the principal functions of the cloverleaf sequence as binding important proteins in the viral lifecycle. These are the viral 3C protease and poly-C binding protein. PCBP is used as a genome circularization element important for viral protein translation and replication. The authors' discussion of the 5'CL is focused on structural biology and biochemistry, and it should include more of a discussion of the virology at play. Particularly, the activities of the 5'CL during viral replication as it relates to VpG addition, uridylation, and minus strand synthesis.

The authors present that the 5' CL structure is in fact an H-type four-way junction. In this organization the sB and sC helices are stacked on top of each other and the sA and sD helices are stacked on top of each other. This makes sense as co-axially stacked helices are commonly found in RNA structures as stacking is an energetically favorable process in RNA folding. The organization of the four-way junction is maintained by backbone and sequence interactions between two stems and an "A-minor" base-triple interaction the sC stem-loop and the sD py-py region.

The residues involved in this interaction are conserved, however mutation of the conserved adenine residue had little effect on 3C protease binding, and in fact the sD region alone is sufficient to bind 3C in the absence of the sC domain, calling into question what the effect of the observed interaction is.

Conclusions:

Overall, the quality of the structural biology work in this paper is high and the authors present an important structure in the field. However, the paper is lacking a connection between the solved structure and the function of the viral RNA. The paper should include more biochemical or virological work to characterize the interactions of the viral RNA with protein co-factors or to support the importance of structural features of the RNA on protein binding or viral fitness/pathogenicity. I would recommend acceptance if some of this work is added to the manuscript.

Not all of these experiments need to be done, but at least one of these three sets of experiments (or similar) should be required.

- 1) The authors present that replacement of the sD loop sequence abrogates, but does not eliminate, 3C protein binding, implying there are other contacts between the 3C protein and the cloverleaf. This makes sense in the context of the unusually high degree of sequence conservation within the cloverleaf beyond what would be necessary to maintain the cloverleaf secondary structure. More biochemical work could be done to probe the interactions between the 3C protein and the RNA. This could include

RNase or chemical protection mapping of a 5' labeled RNA construct to identify a binding footprint. More use of the ITC binding assay or EMSA could also be applied to study more carefully the effects of base substitutions within the cloverleaf structure on 3C protein binding, particularly by substituting the py-py base pairs with Watson-crick pairs, or replacing the identified sD bulge sequence (A50 and U51).

2) The authors propose that the function of the four-way junction is to separate the PCBP and 3C protein binding sites. This is an intriguing possibility, but untested. If the other binding partner PCBP could be made recombinantly this could be used to test this hypothesis. If this protein cannot be made recombinantly, the use of another of these suggested experiments would be sufficient to address concerns about supporting the relevance of the structure. There are also some reports of viral protein 3AB binding some but not all cloverleaf structures. This could also be investigated.

3) As an important viral replication element, the expectation is that mutations that perturb the structure of the cloverleaf would have effects on viral fitness. Measurement of enterovirus replication in a cell-free or cell culture system could confirm the importance of the identified structural elements. These could be measured by viability assays, one-step growth curves, RNA stability assays, or plaque assays. I recognize that the specialty of this lab is most likely to be structural biology/biochemistry and they may not have the capabilities for some of this research except in collaboration. However, if they cannot provide additional biochemical context for the importance of the identified structural elements, virology methods could be an important way of boosting the paper.

Minor comments:

Figure 2A – only one of the nucleotides in the AUU triloop is proposed to interact with the py-py region, and yet all are implied by the drawn secondary structure.

Lines 185-188 – Paper references similarity to previous chemical probing results on the secondary structure of the RNA. “Well-consistent” is a subjective description and difficult to evaluate, an objective description of how closely the structure matches chemical probing results, referring to specific secondary structure elements or residues would be better.

Lines 189-190 – States that the structure is nearly identical to NMR structures and cites figure 1b. Figure 1b does not show any information on the previous NMR structure.

Lines 271-275 - The authors state that the residue A50 is a crystal contact, then claim that its strong density supports a stable conformation of this bulge. Crystal contacts by nature will have strong electron density. While the point stands that A50 being flipped out likely represents a sampled conformation of

the bulge (further supported by the ability of the U49 residue to make contacts with other residues in the region), the existence of strong density at this position is not a convincing argument.

Lines 284 – 287 – The authors assert that the py-py base pairing stretch does not cause a significant deformation from a standard A-form helix. I would not be so quick to dismiss the effects that even subtle deformations may have on the overall structure of the RNA. A small deformation in the middle of the helix may cause larger changes in the relative positions of the ends. Perhaps a figure with an overlay of the sD stem compared against an ideal A-form helix would be more convincing.

Lines 353-356: Replacement of the sD loop reduces binding of 3C to the cloverleaf structure, however it does not eliminate it. To me this suggests there could be multiple points of interaction with the RNA.

Lines 360-363: The authors claim that since mutating G66 has a significant effect on 3C binding, that this implies that 3C recognizes a UNCG tetraloop. To me it seems like this supports a sequence-specific interaction, but does not necessarily speak to the conformation of the loop.

Lines 384-389 and Supplemental figure 11: I am not familiar enough with this technique in order to be able to determine whether the NMR spectra were assigned correctly. I am assuming that this technique isotopically labels adenine and uridine at the indicated positions in order to simplify NMR assignment? Perhaps a reference to another paper using this technique would clarify here.

Lines 408-418 and figure 5a: Caution should be taken when interpreting this consensus structure. The consensus sequence is extremely sequence-conserved. As such, it cannot show base pair co-variation (if a G is mutated to an A then its complementary C is mutated to a U for example) because there is not enough variation to see this. Therefore it does not necessarily support a specific secondary structure. That being said plenty of evidence suggests that the secondary structure is correct.

More importantly, the authors imply that the high conservation of the A40 residue in the triloop is indicative of a specific structural role, however a number of residues in the structure appear to have similar levels of conservation, and these residues do not appear to have any sequence-specific role in the structure. The question arises: why are these other bases so conserved if their function is only to set up a specific secondary structure?

A deeper structural alignment with sequences containing more variation could add interesting information to the consensus sequence to make it easier to interpret which sequences and structural features are most important. Adding more sequences into an R-scape alignment past what is obtained from R-fam could be valuable here. R-scape tools are available online. A larger library of 5'CL sequences

could be taken from a larger list of distinct enteroviruses or picornaviruses on NCBI, or could be searched on a database using tools such as Infernal (<http://eddylab.org/infernal>).

Lines 493-494: The assertion that the lengths of the P3 helix and P4 helix are conserved is unsupported. Please include a supplementary figure to rigorously support this.

A point-by-point response to the reviewers' comments

Reviewer #1 (Remarks to the Author)

The authors present a crystal structure and additional analysis of a (mutated) 5' RNA clover leaf from the start of the Coxsackievirus B3 genome, in complex with a Fab fragment. This cloverleaf is central to the genomic replication of the virus.

The three most interesting results are (i) the overall architecture of the clover leaf, with SA and SD coaxial, and SB and SC coaxial, and these two long helices parallel to each other and in close contact, forming a compact structure.

This architecture has some similarities but some contrast with a previous NMR/SAXS-based study of a similar region from RV14. The similarity is that SB and SD are parallel but not coaxial to each other in both studies. However, in the NMR/SAXS study, SB and SD are in close contact, while in the present study, SB and SD are at opposite ends of the structure and, in fact, are anti-parallel to each other rather than parallel. The authors suggest that this finding may be related to the 2nd very interesting result, (ii) the observation of interactions between the pyrimidine-pyrimidine base pair region in the center of SD and the loop region at the end of SC. The presence of SD-SC interactions, and therefore the overall clover leaf architecture, is further supported by NMR analysis of NOEs. The third interesting point is that (iii) this compact coaxial structure was observed even in the absence of magnesium, which contrasts with the marked magnesium dependence seen in the RV14 study.

This is an interesting study worthy of publication once the following items are addressed:

Authors: We are thankful to the reviewer for the positive feedback. Responses to the specific comments and issues raised by the reviewer are provided in the following.

Main comments

While some differences between the (present) Coxsackievirus structure and the previous rhinovirus structure can (and are) ascribed to differences in techniques of X-ray crystallography vs. NMR/SAXS, and some to the different sequences of the RNA molecules (which was also discussed by the authors), the following additional factors should also be discussed:

- (i) The current study utilizes a mutant RNA molecule (5'CL2 G66C) in which the loop at the end of SB has been mutated to be complementary to a Fab fragment required for co-crystallization, and the loop on the end of SD was also mutated to improve crystallization results.
- (ii) The presence of the Fab fragment in the co-crystal
- (iii) Crystal contacts, in general
- (iv) In particular, crystal contact between A50, near the four-helix junction region, and a symmetry-related Fab fragment

The major change needed is an earnest discussion of the role that the above factors may play in affecting what may be delicately balanced conformational stabilities. This discussion is necessary

in order to properly assess the differences between the current CVB3 structure and the previous RV14 5'CL structure. For instance, interactions involving loop C are proposed to be critical for the observed conformation, but loop B and loop D have been mutated, so there is potential for a change of interactions. How can it be known that interactions with loop B and loop D are not important? A related minor point: discussion of how the G66C mutation was identified as assisting crystallization would be helpful.

Authors: We thank the reviewer for bringing up these important points. The revised manuscript includes a discussion of these points throughout the manuscript and supplementary information as appropriate.

We have included Supplementary Figure 8 (also shown below) with a detailed discussion of all RNA-Fab crystal contacts in the figure legend. Briefly, the Fab BL3-6 in the Fab-RNA co-crystal appeared bound to its cognate motif grafted in the RNA as the L2 loop, consistent with several other RNA-BL3-6 complex crystal structures reported previously. Other Fab-RNA contacts in the crystal lattice mainly involve the interactions of the side chains of positively charged amino acids, the lysines, and arginines, from the Fab scaffold (other than the CDR regions) with the backbone of the RNA (negatively charged surface), indicating that Fab facilitated the crystal packing by minimizing the electrostatic repulsion between the RNA molecules. Additionally, as shown by our gel-electrophoresis results (Supplementary Figure 3), only the epitope-containing RNA constructs bind with the Fab with a ~ 1:1 ratio giving a single band of Fab-RNA complex, suggesting that these Fab-RNA interactions observed in the crystal lattice are likely the crystal contacts rather than specific binding interactions in the solution.

We agree with the reviewer that loops B and D are essential for the function of enteroviral 5'CLs as these are the binding sites for PCBP and 3C proteins, respectively. However, our results and previous in-solution studies suggest that these loops are not involved in the tertiary interactions. First, previous biochemical probing (with CMCT, DMS, and Kethoxal) and SHAPE results have demonstrated that sB and sD loops are both highly-modified and solvent-exposed, suggesting that they are less likely to be involved in tertiary interactions within the RNA structures. Second, our 3C binding results with the intact 5'CL and isolated sD subdomain showed that an isolated sD stem-loop binds the 3C with similar affinity to an intact 5'CL, supporting that the sD subdomain is required and sufficient to bind the 3C. These results also suggest that it is less likely to be involved in any long-range interactions to maintain the 3C binding competent RNA structure (see new Figure 4 and Supplementary Figures 13, 14, and 15). Third, our new PCBP2 binding results (see Supplementary Figure 18 and also the response to reviewer 2 for details) suggest that the replacement of the sB loop by Fab binding sequence abrogates the human PCBP2 binding to the 5'CL, whereas the sD loop replacement had no effects on the PCBP2 binding, consistent with previous studies that PCBP2 interacts with the sB loop. Interestingly, the binding of PCBP2 with the 5'CL was much stronger with a construct having the C-rich spacer sequence at the 3'-end of the 5'CL (see Supplementary Figure 18 and also the response to reviewer 2 for details), supporting our hypothesis that the function of the observed H-shaped structure is to separate sB loop and sD loop, placing the sB loop and the spacer sequence side-by-side for PCBP2 binding and the sD loop at the opposite end for the 3C binding.

Supplementary Figure 8. The Fab-RNA contacts as observed in the crystals of CVB3 5'CL2a RNA in complex with Fab BL3-6. The scaffold residues from the (a) heavy chain (HC) and (b) the light chain (LC) of the symmetry-related Fab make contact with the RNA. Especially the side chains of positively charged residues, such as lysines and arginines, interact with the RNA backbone. The A50 nucleobase within the sD bulge (a) and A64 nucleotide within the sD tetraloop (d) also make contact with the heavy chain residues of the symmetry-related Fab molecules. The Fab residues and the RNA regions involved in the crystal contacts are highlighted in yellow and red, respectively. The dashed lines represent the hydrogen bonding distances.

Our NMR experiments with and without the Fab chaperone in the context of an unmutated D loop support the presence of long-range interactions between A40 and the Py-Py helix, suggesting that our crystal structure represents its solution conformation. Given the high conservation of sequence and secondary structures among enteroviral 5'CL (Figure 5a and new Supplementary Figures 19, 20, and 21; also shown below in response to reviewer 2), it is plausible that RVB14 5'CL will also fold into the CVB3-like structure. Nevertheless, the previous NMR and SAXS-based studies have shown different behavior of RVB14 5'CL, such as magnesium-dependent conformation. We are also pursuing structural studies with the RVB14, and the findings will be reported in due course.

Supplementary Figure 4. Models of the sD tetraloops for (a) uCACGg (wild-type) and (b) uCACCCg (crystallized construct) as predicted by the ROSIE (Rosetta Online Server that Includes Everyone, https://rosie.rosettacommons.org/rna_denovo).^{1, 2} The wild-type sequence folded into a UNCG-type tetraloop, consistent with the previous in-solution NMR results,³⁻⁵ whereas the mutant folded into a GNRA-type tetraloop similar to that observed in the wild-type UNCG-type tetra-loop crystal structure. Noticeably, nucleotide A64 that facilitated the crystal contacts in the crystallized loop (see Supplementary Figure 8

below) appeared extended out in the wild-type loop that would have clashed with symmetry-related Fabs, which perhaps explains the low-resolution X-ray diffraction of the 5'CL2 construct with uCACGg loop compared to the mutant, 5'CL2a construct with the uCACCg loop.

The G66C mutation was chosen based on computational modeling of the sD loop mutants. For each L2 loop mutant, we predicted the sD hairpin models using the ROSIE (Rosetta Online Server that Includes Everyone, https://rosie.rosettacommons.org/rna_denovo) with particular attention to the L2 tetraloop conformation. The G66C mutant folded into a GNRA-type tetraloop compared to a UNCG-type tetraloop for the wild-type L2. As the GNRA-type tetraloops are known to facilitate crystal contacts, we used this G66C mutation in the context of the 5'CL2, which yielded robust crystals in complex with the Fab that diffracted to 1.9 Å resolution. Consistent with this prediction, the L2 loop in the crystal folded into a GNRA-type tetraloop that made crystal contacts with symmetry-related Fab molecules. However, in the wild-type loop that folds into UNCG-type tetraloop (consistent with previous NMR and our computational model), the G66 is flipped out, which clashes with symmetry-related Fabs. It perhaps explains the low-resolution diffraction of the 5'CL2 (~8 Å) compared to the G66C mutant, 5'CL2a (1.9 Å resolution). We have discussed these points in the revised manuscript as appropriate and included Supplementary Figure 4 (also shown above) to discuss the modeled tetraloop conformations.

Other comments/questions:

Line #213-228

discussion of the widened groove in the rhinovirus SB region should focus on the smaller number of base pairs in rhinovirus SB. This is mentioned later on in the paragraph, but it should be the main point, and thus the CVB3 SLB result does not present a substantial difference from the RV14 SLB result. Or at least the difference is due to a different number of base pairs rather than different techniques, etc.

Authors: We have revised this discussion according to the reviewer's suggestion.

...Although it is possible that the differences in major groove widths for CVB3 and RVB14 sB helix reflect the differences in methodologies used for these RNA structure determination,⁶ the structural differences between our CVB3 and RVB14 sB helices are likely because of the different number of base-pairs flanking the 4-way junction that constitute the sB helix (Supplementary Figure 10). The nine base-paired sB helix in the CBV3 sB subdomain forms a near complete helical turn compared to a slightly over a half helical turn formed by the seven base-paired sB helix in the RVB14. Additionally, compared to all canonical base-pairs in the CVB3 sB, the RVB14 sB helix contains a non-canonical G•U pair (Supplementary Figure 10)...

362-364

The authors suggest that the minimal effect of magnesium on binding to 3C protease provides evidence that the CVB3 conformational architecture has no dependency on magnesium. I do not see that this follows since all evidence points to 3C protease interacting only with SD and its loop, an interaction which would not manifestly be dependent on tertiary structure (relative alignment

of helices). Of course, the authors provide other evidence that CVB3 does not seem to show this magnesium dependence, but 3C binding is not a convincing part of this evidence.

Authors: We agree with the reviewer's comment. The revised manuscript discusses this point according to the reviewer's suggestion.

...The divalent cations such as Mg^{2+} are known to stabilize the RNA tertiary structures. Nevertheless, the presence or absence of Mg^{2+} in the solution did not alter the binding affinities of 3C^{pro} with the 5'CL constructs significantly (for the 5'CL2, apparent K_d with 10 mM Mg^{2+} = $1.4 \pm 0.2 \mu M$ compared to $1.30 \pm 0.09 \mu M$ in an Mg^{2+} dialyzed solution), supporting that tertiary structures, compared to the secondary structural features, within the 5'CL may have no major effect on 3C^{pro} binding...

368

Reference is made to 2D NMR data in Supp. Fig. 10, but this Fig. is only a small region of a 1D NMR spectrum. This argument would be far more convincing with the 2D data of the imino proton region, either a NOESY or a 15N-HSQC.

Authors: We thank the reviewer for this suggestion. We added two regions of a 2D dataset of the measured U^{6R} sample (Supplementary Figure 17). Additionally, we adjusted the main text to reflect this change. The 2D data show only small chemical shift perturbations, which is typical for changes in salt composition. The larger CSPs, or intensity changes that would suggest structural changes, were not observed.

401-407

If the purine-purine base pairs are needed to position SD alongside SC, and thus to position SD and SB in an anti-parallel fashion, then why are the purine-purine base pairs dispensable for (-)-strand synthesis, which uses this structure as a platform?

Authors: We appreciate the reviewer's comment on this point. We have discussed these points more thoroughly in the revised manuscript and supplementary information as appropriate.

The replication process of enteroviral genomes involves two distinct steps. First, the synthesis of (-)-strand RNA using the genomic (+)-strand RNA as the template, and second, back synthesis of the (+)-strand genomic RNA using the newly made (-)-strand RNA as a template. Previous studies using cell-free assays have shown that replacing Py-Py mismatch with Watson-Crick base pairs does not affect (-)-strand synthesis but abolishes the (+)-strand synthesis, indicating that the Py-Py region in the sD subdomain is perhaps more important for the back synthesis of the (+)-strands using (-)-strands as the templates.

Structurally, the integrity of three Py-Py base pairs to form an A-form helix appears more critical than nucleotide identities that preserve the overall structure of the sD. It means any compensatory mutations that do not alter the A-form helicity would still permit the A-minor type interactions between the A40 and Py-Py helix (see new Supplementary Figure 11 and response to reviewer 2 for details), the replacement of Py-Py mismatch with complementary base pairs would still position the sD alongside sC and thus separates 3C and PCBP binding sites (the sD

and sB subdomain, respectively) in an antiparallel fashion. Nevertheless, the deletion of a nucleotide from the Py-Py region, which would change the helicity and relative positions of the sD, reduced (-)-strand synthesis, suggesting that the positioning and integrity of these subdomains are important for the (-)-strand synthesis. Moreover, consistent with previous studies, our 3C binding results suggest that the A40-Py-Py interactions do not impact 3C binding as isolated sD binds the 3C with similar affinity as the intact 5'CL (see new data in Figure 4c and Supplementary Figures 14 and 15), leading us to hypothesize that A40-Py-Py long-range interactions are to position the PCBP binding sB loop close to the spacer region between the 5'CL and IRES. Consistent with previous biochemical studies showing the binding of PCBP2 with both the sB loop and the spacer region, our preliminary binding results with the recombinant PCBP2 support this hypothesis (see Supplementary Figure 18 and response to reviewer 2 for details). A thorough study of PCBP2 binding with multiple enteroviral 5'CLs is underway, and the results will be reported in due course.

Although the effects of A40 mutations in (-)- and (+)-strand synthesis are yet to be investigated, absolute conservation of the A40 against intense selection pressure underscores its requirement to maintain A40-Py-Py tertiary interactions for viral genome replication. On the other hand, this scenario is less likely to be the same in an analogous cloverleaf secondary structure formed in the 3'end (3'CL) of the newly synthesized (-) strand. The structure is proposed to function analogously to the 5'CL, but the 3'CL binds a different set of proteins compared to the 5'CL. For example, the viral 2C protein has been shown to bind the 3'CL, not the 5'CL. It is beyond the scope of this study to investigate the 3'CL structures, but perhaps the requirement for the Py-Py region in the 5'CL is functionally more prominent in the (+)-strand synthesis using the 3'CL of the (-)-strand RNA template.

Typos

Line #60 consensus

Authors: We have fixed this typo in the revised manuscript.

264 check Figure reference is correct

Authors: Thank you for catching this error. It has been fixed in the revised manuscript.

Reviewer #2 (Remarks to the Author)

Summary:

Das et al present a crystal structure of the 5' cloverleaf sequence from coxsackievirus B3, an important enterovirus which also serves as a model system for several other medically important viruses, including poliovirus and hepatitis A. The 5' cloverleaf sequence is an essential RNA element within enteroviruses necessary for viral replication and pathogenicity.

The authors used a strategy of including a Fab binding region into their RNA construct to favor the formation of crystal contacts. Importantly, they demonstrate that the location of their Fab binding site does not disrupt the enterovirus protein binding activity of the RNA. Another single-point mutation was made, which increased resolution. This point mutation did affect protein binding; however, an analysis of the crystal structure would suggest that it is unlikely to cause a global change in RNA structure that would affect the overall conclusions of the paper.

The resolution of the solved structure is high (1.9Å), permitting the unambiguous assignment of electron density. The authors report only a single RNA-RNA crystal contact suggesting that the observed structure should lack crystal artifacts, however, the paper should include a discussion of protein-RNA crystal contacts in order to confirm that they are not perturbing the global RNA structure. Supplementary figure 5 is a useful starting point; however, there should be an inset to show all other significant crystal contacts within the RNA structure. It would also be useful to include a color map of b-factors somewhere in the supplement.

Authors: We are thankful to the reviewer for the critical comments and constructive suggestions. We have now included Supplementary Figure 8 (also shown above in response to reviewer 1) with a detailed discussion of all RNA-Fab crystal contacts. The revised manuscript also includes a color map of crystallographic B-factors as Supplementary Figure 6 (also shown below). These changes have been discussed in the revised manuscript as appropriate.

The authors present the principal functions of the cloverleaf sequence as binding important proteins in the viral lifecycle. These are the viral 3C protease and poly-C binding protein. PCBP is used as a genome circularization element important for viral protein translation and replication. The authors' discussion of the 5'CL is focused on structural biology and biochemistry, and it should include more of a discussion of the virology at play. Particularly the activities of the 5'CL during viral replication as it relates to VPg addition, uridylation, and minus-strand synthesis.

Authors: We thank the reviewer for this suggestion. In the revised manuscript, we have added a concise discussion of the 5'CL and its role from the virological and overall genome replication point of view. We have added these points throughout the manuscript as appropriate. However, the majority of the points are included in the second paragraph of the "INTRODUCTION" section and the first paragraph of the "DISCUSSION" section.

...A stem-loop RNA domain, *cre*, within the 2C coding region of the viral genome also interacts with the 3CD,⁷⁻⁹ promoting the uridylation of the viral VPg protein,^{8,10} which afterward serves as a primer for the (-)-strand RNA synthesis by the RdRp D.^{7,11} Additionally, the sequence and structural integrity of the 5'CLs

have also been shown to influence the VPg uridylation and genomic stability, underscoring multiple critical roles of RNA structural features within the enteroviral 5'CLs.^{12, 13}...

...The 5'CL is one of the major components of the enteroviral genome replication machinery that directly binds viral and cellular protein factors to form a replication-competent RNP complex. The sD subdomain of 5'CL interacts with viral 3CD fusion protein through its 3C^{pro} that brings the viral polymerase D^{pol} to the replication initiation site. The sB subdomain binds PCBP2, which then interacts with PABP-poly(A) tail complex at the 3'end to circularize the viral genome. The 3CD also interacts with *cre*, a stem-loop RNA domain located within the 2C coding region, to promote the uridylation of viral VPg protein. The resulting VPg-pUpU then serves as a primer for the (-)-strand RNA synthesis by active D^{pol} polymerase, an autocleavage product of the 3CD precursor. Additionally, the conserved RNA sequence and structural features in the 5'CL influence the VPg uridylation and viral genome stability and may facilitate the replication machinery to recognize the enteroviral genome in a milieu of myriad cellular RNAs.¹⁴...

Supplementary Figure 6. The crystal structure of the 5'CL2a in complex with Fab BL3-6 colored according to the crystallographic B-factors. The gradient from blue to red indicates the lowest (29 Å²) and the highest (130 Å²) B-factors for the structure.

The authors present that the 5' CL structure is, in fact, an H-type four-way junction. In this organization, the sB and sC helices are stacked on top of each other, and the sA and sD helices are stacked on top of each other. This makes sense as co-axially stacked helices are commonly found in RNA structures, as stacking is an energetically favorable process in RNA folding. The organization of the four-way junction is maintained by backbone and sequence interactions

between two stems and an “A-minor” base-triple interaction between the sC stem-loop and the sD Py-Py region.

The residues involved in this interaction are conserved, however mutation of the conserved adenine residue had little effect on 3C protease binding, and in fact, the sD region alone is sufficient to bind 3C in the absence of the sC domain, calling into question what the effect of the observed interaction is.

Authors: We agree with the reviewer that A40U mutation has little effect on the 3C binding. With additional binding experiments using ITC with the mutant constructs, including the isolated sD subdomain, intact 5'CL with canonical base pairs in the sD helix and sD bulge mutants, it is more obvious that the sD subdomain is sufficient for binding the 3C protein (discussed below in more details). We hypothesize that the observed long-range A40-Py-Py interactions are required beyond 3C binding. First, during replication, the 5'CL recruits the 3CD fusion protein (D is the RNA-dependent RNA polymerase), and thus, these RNA interactions may be needed for interactions with the polymerase. Second, the PCBP binding occurs with the sB loop and the spacer region between the 5'CL and the IRES domains. The A40-Py-Py interactions stabilize the H-shape structure of the 5'CL to position the sB loop and the spacer region adjacent to each other for binding PCBP protein. The mutation to A40 perhaps separates these two PCBP binding sites, affecting the affinity of PCBP binding rather than the 3C binding. We are currently working on testing these hypotheses through binding studies of recombinantly expressed human PCBP2 and CVB3 3CD fusion proteins. We have not obtained conclusive data due to the technical challenges associated with the expression of these proteins. However, our preliminary results of the PCBP2 binding using native gel electrophoresis for the 5'CL and the longer RNA constructs with a C-rich spacer suggest that the H-shaped 5'CL structure stabilized by the A40-Py-Py long-range contact juxtaposes the PCBP2 binding sites, the sB loop, and the spacer well-separated from the 3Cpro binding site, the sD subdomain (discussed below in more details).

Additionally, we have determined the crystal structures of Rhinovirus B and Rhinovirus C using similar approaches, but the Fab binding motif grafted in the sD loop. Interestingly, despite different Fab binding positions, the topology and the A40-Py-Py interactions in these structures are identical to the CVB3 5'CL reported here, supporting that these interactions are not the artifacts of crystallization of a particular RNA, and they represent a conserved tertiary motif among enteroviral 5'CLs. These studies are beyond the scope of the current manuscript and will be reported separately.

Conclusions:

Overall, the quality of the structural biology work in this paper is high, and the authors present an important structure in the field. However, the paper is lacking a connection between the solved structure and the function of the viral RNA. The paper should include more biochemical or virological work to characterize the interactions of the viral RNA with protein co-factors or to support the importance of structural features of the RNA on protein binding or viral fitness/pathogenicity. I would recommend acceptance if some of this work is added to the manuscript.

Not all of these experiments need to be done, but at least one of these three sets of experiments (or similar) should be required.

1) The authors present that replacement of the sD loop sequence abrogates but does not eliminate 3C protein binding, implying there are other contacts between the 3C protein and the cloverleaf. This makes sense in the context of the unusually high degree of sequence conservation within the cloverleaf beyond what would be necessary to maintain the cloverleaf secondary structure. More biochemical work could be done to probe the interactions between the 3C protein and the RNA. This could include RNase or chemical protection mapping of a 5' labeled RNA construct to identify a binding footprint. More use of the ITC binding assay or EMSA could also be applied to study more carefully the effects of base substitutions within the cloverleaf structure on 3C protein binding, particularly by substituting the Py-Py base pairs with Watson-crick pairs or replacing the identified sD bulge sequence (A50 and U51).

Authors: We appreciate the reviewer for suggesting these experiments. We have now performed 3C binding experiments using ITC for several new RNA constructs, including isolated sD mutants, intact 5'CL with canonical base pairing in the sD helix, and the sD bulge mutants. The revised manuscript and the supplementary information present and discuss these results as appropriate.

...Although the sD loop mutants showed a reduced affinity to 3C^{pro}, the observable affinities of these mutants suggest additional contacts of 3C^{pro} with the 5'CL. To test this hypothesis, we performed 3C^{pro} binding studies with the dinucleotide bulge and Py-Py mutants. The elimination of the dinucleotide bulge by adding the complementary nucleotides (5'CL-sD-NB, Supplementary Figure 14) showed ~ seven times less affinity (apparent $K_d = 10.1 \pm 2 \mu\text{M}$, Figure 4c) with the 3C^{pro} compared to the WT 5'CL (apparent $K_d = 1.4 \pm 0.14 \mu\text{M}$, Supplementary Figure 14), indicating direct contacts of the bulge nucleotides with the 3C^{pro}. However, a construct (5'CL-sD-CN, Supplementary Figure 14) with Py-Py region replaced by the canonical base pairs displays a similar affinity (apparent $K_d = 3.4 \pm 0.7 \mu\text{M}$, Figure 4c) as the WT 5'CL, supporting that Py-Py region is less likely to be involved in the 3C^{pro} binding directly. Moreover, consistent with the previous reports,¹⁵ an isolated sD subdomain (sDi, Supplementary Figure 15) binds 3C^{pro} with a similar affinity (apparent $K_d = 2.1 \pm 0.12 \mu\text{M}$, Figure 4c) as the intact WT 5'CL, suggesting that the sD subdomain is sufficient for 3C^{pro} binding through specific interactions of the sD loop and sD bulge. Furthermore, 3C^{pro} binding studies with isolated sD constructs with the canonical base pairing of both U49A50 (sDi-NB, apparent $K_d = 5 \pm 0.1 \mu\text{M}$), U49 only (sDi-U49P, apparent $K_d = 4.9 \pm 0.87 \mu\text{M}$), or A50 only (sDi-A50P, apparent $K_d = 3.5 \pm 0.6 \mu\text{M}$) indicate that the overall structure of the bulge is critical for 3C^{pro} binding rather than the identity of the bulged nucleotide (see Supplementary Figure 15 for the constructs and ITC data). These results are consistent with two different conformations of the bulge being observed in the previous NMR⁴ and our crystal structures, and *in vitro* replication studies showing deletion of both U49A50 but not either U49 or A50 or swapping their positions abrogated the (-)-strand synthesis.¹³

¹⁵...

Figure 4 Structure of CVB3 5'CL sD subdomain and its binding interactions with 3C protein. (a) Representative ITC profile for the WT 5'CL. The heat is released upon successive injections of 2 μ l of CVB3 3C ($\sim 400 \mu$ M) to the RNA solution ($\sim 10 \mu$ M) in the calorimetry cell. (b, c) The binding curves from the ITC data for the binding of the 3C with various crystallization RNA constructs (see Supplementary Figures 13, 14, and 15 for details). (d) Structure of the CVB3 5'CL2a, showing the tertiary interactions between the sC and sD subdomains, especially the docking of the sC-loop A40 into the Py-Py region of the P4 helix within the sD subdomain. (e) The details of A-minor type tertiary interactions between the A40 and the C56•U73 base-pair within the Py-Py region. (f) An A40U mutation model showing disruption of the A-minor interactions between the sC-loop and the Py-Py helix. (g) The binding curves from the ITC data for the binding of the 3C with the A40U mutant of the parent 5'CL2 construct. Black dashed lines in d, e, and f represent heteroatoms within hydrogen bonding distances. Gray mesh in d and e represents the $2|F_o| - |F_c|$ electron density map at contour level 1σ and carve radius 1.8 \AA .

2) The authors propose that the function of the four-way junction is to separate the PCBP and 3C protein binding sites. This is an intriguing possibility but untested. If the other binding partner PCBP could be made recombinantly this could be used to test this hypothesis. If this protein cannot be made recombinantly, the use of another of these suggested experiments would be sufficient to address concerns about supporting the relevance of the structure. There are also

some reports of viral protein 3AB binding some but not all cloverleaf structures. This could also be investigated.

Authors: Our binding studies with 3C protein strongly suggest that observed A40-Py-Py interaction in the 5'CL crystal structure has little or no effect on 3C binding. However, based on the observed side-by-side positioning of the two proposed PCBP binding sites in the crystal structure due to the stabilization of the four-way junction by A40-Py-Py tertiary interaction, We hypothesized that a more prominent structural role of these interactions is to stabilize the 4-way junction of the 5'CL that positions the 3Cpro binding site (sD loop) away from the host PCBP binding site (sB loop) to avoid potential steric clashes between 3Cpro and PCBP during the replication machinery assembly. To test this hypothesis, we recombinantly expressed and purified the full-length human PCBP2 protein and performed the binding assays using gel electrophoresis (Supplementary Figure 18, also shown below). As PCBP2 has been shown to bind both the sB loop and the C-rich spacer region between the 5'CL and IRES, we also performed binding tests with the longer RNA constructs that contain both the 5'CL and spacer region. Consistent with the previous studies that PCBP2 recognizes the sB loop, we observed that the WT 5'CL but not the sB loop mutant 5'CL2 bind PCBP2. Interestingly, the A40U mutant, which had no effect on 3Cpro binding, interacts with the PCBP2 with much less affinity compared to the WT 5'CL, suggesting that the A40-Py-Py interaction has some roles in PCBP2 binding to the 5'CL structure. Strikingly, the longer RNA construct 5'CLWTSP with the sB loop and spacer sequence binds the PCBP2 tightly compared to the 5'CL without the spacer, indicating that the sB loop and the spacer reside close to each other, which is consistent with the observed juxtaposition of sA and sB subdomains in our crystal structure. Although the effect of A40-Py-Py interaction on the PCBP2 binding and 5'CL function remains to be studied thoroughly, our preliminary results suggest that the H-shaped 5'CL structure stabilized this A40-Py-Py long-range contact juxtapose the PCBP2 binding sites, the sB loop, and the spacer well-separated from the 3Cpro binding site, the sD subdomain.

This discussion has been added in the revised manuscript under the new heading "Potential roles of A40-Py-Py interactions in 3Cpro and PCBP2 proteins binding".

3) As an important viral replication element, the expectation is that mutations that perturb the structure of the cloverleaf would have effects on viral fitness. Measurement of enterovirus replication in a cell-free or cell culture system could confirm the importance of the identified structural elements. These could be measured by viability assays, one-step growth curves, RNA stability assays, or plaque assays. I recognize that the specialty of this lab is most likely to be structural biology/biochemistry, and they may not have the capabilities for some of this research except in collaboration. However, if they cannot provide additional biochemical context for the importance of the identified structural elements, virology methods could be an important way of boosting the paper.

Authors: We agree with the reviewer that it is important to perform structure-guided functional assays to establish the structure-function relationship. Also, we appreciate the reviewer for understanding our current limitations in performing virological measurements. We have

established some collaborations to perform these experiments, and the outcomes will be reported in due course.

Supplementary Figure 18. The human PCBP2 binding tests for the 5'CLWT (a), 5'CLWTA40U (b), 5'CL2 (c), 5'CLWTSP (d), and 5'CLSPA40U (e) (see Methods for the experimental details). An SDS PAGE gel (f) showing the expected length and high purity of the recombinantly expressed human PCBP2 protein (see Methods for the expression and purification details). The nPAGE for testing the binding of PCBP2 with the core CVB3 5'CL constructs as shown in a-c (g) and with the longer constructs as shown in d and e (h). Each lane contains 100 ng of RNA. Although we did not observe a clear RNA-PCBP2 complex band for the core constructs, perhaps due to the weak affinity of PCBP2 for the sB loop, a PCBP2 dose-dependent shift of the RNA bands indicated the complex formation. Nevertheless, the longer RNA constructs with the 5'CL

core plus the C-rich spacer sequence showed sharp, slow-migrating PCBP2-RNA complex bands in a PCBP2 dose-dependent manner, suggesting strong binding. Detailed studies with ITC are underway.

Minor comments:

Figure 2A – only one of the nucleotides in the AUU triloop is proposed to interact with the Py-Py region, and yet all are implied by the drawn secondary structure.

Authors: We apologize for this error. The figure has been modified accordingly.

Lines 185-188 – The paper references similarity to previous chemical probing results on the secondary structure of the RNA. “Well-consistent” is a subjective description and difficult to evaluate, an objective description of how closely the structure matches chemical probing results, referring to specific secondary structure elements or residues, would be better.

Authors: Thanks to the reviewer for this suggestion. We have modified this discussion in the revised manuscript as suggested by the reviewer. Additionally, we have added Supplementary Figure 9 (also shown below) to discuss in detail the previous biochemical probing and SHAPE results and compare these results with our crystal structure.

Lines 189-190 – It states that the structure is nearly identical to NMR structures and cites figure 1b. Figure 1b does not show any information on the previous NMR structure.

Authors: We apologize for this error. We have removed the figure citation and now provided a separate figure (Supplementary Figure 9, also shown above) to compare previous NMR structures of sD and sB subdomains with that observed in our crystal structure.

Supplementary Figure 9. Comparison of previous biochemical probing results and NMR-based structures with the CVB3 5'CL crystal structure. (a) The secondary structure of CVB3 5'CL shows the modified positions (filled squares) by Kethoxal, DMS, or CMCT according to the data taken from Bailey et al.¹⁶ and Prusa et al.¹⁷ The black and gray squares identify strongly and moderately modified positions, respectively. (b) The same CVB3 5'CL structure shows the SHAPE reactivity at each position, as indicated by the white-to-red color, according to data retrieved from Mahmud et al.¹⁸ The nucleotides colored white and dark red refer to the lowest and highest SHAPE reactivity, respectively. The pink-colored nucleotides represent intermediate SHAPE reactivity. Both biochemical probing and SHAPE data shown here were performed in the context of the intact 5'UTR. (c) NMR structure of CVB3 sD (without the bulge portion, PDB: 1RFR),⁵ and (d) a consensus sD (including the bulge portion, PDB: 1TXS).⁴ Except for the flipped nucleotide (A or U) in the bulge, the NMR-derived secondary structure of the isolated CVB3 sD (e) is identical to that derived from the crystal structure of intact 5'CL (f). The NMR structure of HRV-14 sB (g) taken from a publication by Warden et al.¹⁹ shows no unpaired nucleotides within the sB helix. Overall, the secondary structural features of the enteroviral 5'CL observed in the solution by biochemical probing, SHAPE, and

NMR agree with those observed in the crystal, supporting that the crystal structure of CVB3 5'CL represents that in the solution.

Lines 271-275 - The authors state that the residue A50 is a crystal contact, then claim that its strong density supports a stable conformation of this bulge. Crystal contacts, by nature, will have strong electron density. While the point stands that A50 being flipped out likely represents a sampled conformation of the bulge (further supported by the ability of the U49 residue to make contact with other residues in the region), the existence of strong density at this position is not a convincing argument.

Authors: We have revised this discussion according to the reviewer's suggestion. We agree with the reviewer that flipped-out A50 represents a sampled conformation of the bulge, and the crystal contacts with the Fab stabilize this conformation. Nevertheless, previous studies have shown that a single nucleotide bulge is sufficient for the 5'CL function, and the phylogenetic analysis suggests bulged-out A50 is conserved more compared to other nucleotides at the 49th position (U49 for CVB3) among enteroviral 5'CLs (see Supplementary Figures 19, 20, and 21), indicating that flipped-out A50 may be important for interactions with the 5'CL binding proteins. The revised text in the manuscript is as follows.

...While the crystal structure may represent a sampled conformation of the bulge with flipped-out A50 stabilized by the crystal contact, a strong hydrogen bonding network within the bulge supports a stable rather than a dynamic configuration of this bulge. Also, the 49th nucleotide is less conserved compared to A50 among enteroviral 5'CLs, indicating that flipped-out A50 may be important for interactions with the 5'CL binding proteins...

Lines 284 – 287 – The authors assert that the Py-Py base pairing stretch does not cause a significant deformation from a standard A-form helix. I would not be so quick to dismiss the effects that even subtle deformations may have on the overall structure of the RNA. A small deformation in the middle of the helix may cause larger changes in the relative positions of the ends. Perhaps a figure with an overlay of the sD stem compared against an ideal A-form helix would be more convincing.

Authors: Thanks to the reviewer for this suggestion. We further analyzed our sD helix structure and compared it with the corresponding ideal A-form helix. For facile comparison, we created an ideal helix for the same sD sequence but eliminated the bulge and Py-Py mismatch by canonical base pairing. A comparative analysis has been added to the revised manuscript as Supplementary Figure 11 (also shown below).

Supplementary Figure 11. Comparison of the sD subdomain structural features with an ideal A-form RNA helix. (a) Superposition of an ideal A-form helix (yellow) with the sD subdomain (orange) of the intact 5'CL crystal structure (RMSD = 1.848 Å). For simplistic comparison, the ideal A-form helix was created computationally for the same sD sequence, G47 to U62, excluding the loop L4. (b) Superposition of the ideal A-form helix (yellow) with the isolated sD subdomain (orange) without the loop L4 (RMSD = 1.807 Å). As observed in the crystal structure, the A40 interaction with the sD helix is also shown for comparison. Except for the slight widening of the ideal helix compared to the Py-Py helix (c), the sD helix after the bulge showed no significant deviation from ideal A-form helix (RMSD = 1.579 Å). Additionally, the presence of the Py-Py mismatch pair in the middle of the sD helix seems not to change the relative orientations of the sD helix ends (a-c). Interestingly, the replacement of the Py-Py region by canonical base pairs still allows A40-sD helix interactions (d) in a similar fashion as observed in the crystal structure (e), which is consistent with previous observations that these changes did not inhibit the (-)-strand synthesis.¹³

¹⁵ Perhaps the A40-sD helix interactions are to correctly position the 3C and PCBP binding sites at the opposite ends of the 5'CL structure, which is consistent with previous observations that any deletions in the Py-Py region, which likely bend the sD helix in the middle changing the relative positions of 3C and PCBP binding sites, inhibits the (-)-strand synthesis. Absolute conservation of the Py-Py region may be required for the back-synthesis of the (+)-strand using the (-)-strand as the template, where a similar cloverleaf structure (3'CL) is likely to form but with the complementary sequence, consistent with the ability of 3'CL to bind a different set of proteins compared to the 5'CL.²⁰⁻²⁴ Moreover, although Py-Py non-canonical sD helix compared to the corresponding ideal A-form helix preserved the orientation of the ends and interactions with the A40, the presence of the bulge at the base of the sD helix seems to shorten the length of the sD helix between the four-way junction and the Py-Py region (f). The arrangement helps precisely position the central C56•U73 pair of the Py-Py helix for interactions with the A40, consistent with the previous observations that the sD bulge is essential for the (-)-strand synthesis.^{13, 15}

Lines 353-356: Replacement of the sD loop reduces the binding of 3C to the cloverleaf structure, however it does not eliminate it. To me, this suggests there could be multiple points of interaction with the RNA.

Authors: Thanks to the reviewer for this suggestion. We agree with the reviewer that the replacement of the sD loop by the Fab binding loop does not eliminate the binding of 3C with the 5'CL RNA, suggesting that replaced pentaloop itself may interact with the 3C non-specifically or there could be other interactions within the RNA structure. Our new results for binding of 3C with Py-Py, the bulge, and isolated sD mutants support that the sD dinucleotide bulge is also important for binding the 3C (Figure 4c and Supplementary Figures 14 and 15). Yet, it is possible that the grafted pentaloop also interacts with the 3C non-specifically, consistent with the previous observations that replacement of the sD tetraloop by the CAG triloop inhibited (-)-strand synthesis but not the CACCG pentaloop. We are pursuing similar binding studies with rhinovirus 5'CL, which has triloop in the sD loop, to understand the details of this binding, which will be reported in due course. Appreciating the reviewer's suggestion, we have modified the related discussions in the revised manuscript with new results for 3C binding studies.

Lines 360-363: The authors claim that since mutating G66 has a significant effect on 3C binding that this implies that 3C recognizes a UNCG tetraloop. To me, it seems like this supports a sequence-specific interaction but does not necessarily speak to the conformation of the loop.

Authors: A comparative analysis of sD tetraloop conformations of the CVB3 crystal structure, computational models (see Supplementary Figure 4 and response to reviewer 1), and previously reported NMR structures suggest that the wildtype CACG sequence adopts a UNCG-type tetraloop, whereas the mutated CACC sequence likely folds into a GNRA-type tetraloop. We agree with the reviewer that we cannot rule out the sequence-specific effects. However, the relative position of G66 in the UNCG tetraloop and C66 in the GNRA tetraloop is similar, and it is the A64 that adopts different conformation in these tetraloops, leading us to hypothesize that the tetraloop confirmation plays a key role in binding 3C. Appreciating the reviewer's suggestion, we have slightly modified this discussion in the revised manuscript.

... However, the relative positioning of G66 in the UNCG-type and C66 in the GNRA-type tetraloop is similar (Supplementary Figure 4), indicating that the tetraloop conformation plays an important role in 3C^{pro} binding beyond the sequence-specific interactions. Taken together, these analyses support that 3C^{pro} recognizes a UNCG-type tetraloop within the sD loop, and any mutations that cause alterations to this structure abrogate 3C^{pro} binding with the enteroviral 5'CLs...

Lines 384-389 and Supplemental figure 11: I am not familiar enough with this technique in order to be able to determine whether the NMR spectra were assigned correctly. I am assuming that this technique isotopically labels adenine and uridine at the indicated positions in order to simplify NMR assignment. Perhaps a reference to another paper using this technique would clarify here.

Authors: We added two additional references (Du et al., 2003; Du et al., 2004) from previous assignments of parts of the RNA that was analyzed here. These assignments were used to transfer and cross-validate some of the assignments that were done in this study (see method section “NMR data acquisition, processing, and analysis.”)

Furthermore, two references were added, one describing the assignment strategy in more detail (Kotar et al., 2020) and another one leading as an example paper that pioneered this assignment strategy (Keane et al., 2015).

Lines 408-418 and figure 5a: Caution should be taken when interpreting this consensus structure. The consensus sequence is extremely sequence-conserved. As such, it cannot show base pair co-variation (if a G is mutated to an A, then its complementary C is mutated to a U, for example) because there is not enough variation to see this. Therefore, it does not necessarily support a specific secondary structure. That being said, plenty of evidence suggests that the secondary structure is correct.

Authors: We agree with the reviewer. We have now analyzed about 5000 enteroviral sequences available in the NCBI database. The new analysis (see below) covers sequences from enterovirus A to D species and rhinovirus A to C species to include more variability in sequences.

More importantly, the authors imply that the high conservation of the A40 residue in the triloop is indicative of a specific structural role, however, a number of residues in the structure appear to have similar levels of conservation, and these residues do not appear to have any sequence-specific role in the structure. The question arises: why are these other bases so conserved if their function is only to set up a specific secondary structure?

Authors: We thank the reviewer for raising this concern. Based on 3C binding experiments, which support that the A40 tertiary interaction has less influence on 3C binding, we hypothesize that it perhaps has a structural role in stabilizing the H-shape conformation of the 5'CL that separates the 3C and PCBP binding sites to position them at the opposite ends. Our new PCBP binding experiments, as discussed above (Supplementary Figure 18), are consistent with this hypothesis. It is noteworthy that the 5'CL assembles several other factors, such as viral polymerase D (binds to the 5'CL as 3CD fusion) and VPg, and specific sequence conservation might be needed for interactions with those factors. Moreover, the sequence conservation in the 5'CL may reflect

sequence and structural requirements in the 3'CL when (+)-strand synthesis occurs using other viral and host protein factors. For example, previous studies have shown that the 3'end UGUUUU sequence of poliovirus 3'CL in the negative strand binds viral 2C protein, but the protein does not bind the same sequence present in the 5'end of the 5'CL,²¹ although the secondary structure (the sA stem) is same in both cases.

A deeper structural alignment with sequences containing more variation could add interesting information to the consensus sequence to make it easier to interpret which sequences and structural features are most important. Adding more sequences into an R-scape alignment past what is obtained from R-fam could be valuable here. R-scape tools are available online. A larger library of 5'CL sequences could be taken from a larger list of distinct enteroviruses or picornaviruses on NCBI or could be searched on a database using tools such as Infernal (<http://eddylab.org/infernal>).

Authors: We thank the reviewer for this suggestion. We have now performed more rigorous bioinformatics and sequence analysis alignment using over 5000 enteroviral sequences available in the NCBI database. The results with associated figures are discussed as appropriate in the revised manuscript and the supplementary information (also shown below).

Supplementary Figure 19. The consensus sequences for the 7 types of human enterovirus species belonging to the enterovirus genus. The sequences available in the NCBI database nucleotide collection (nt/rt) (<https://blast.ncbi.nlm.nih.gov/>) for human enteroviral species were filtered for coverage completeness of the 5'CL region within the 5'UTR. These seed sequence data were then aligned using MuscleWS (<http://www.compbio.dundee.ac.uk/jabaws/>)²⁵ and visualized in Jalview (<https://www.jalview.org/>) software.²⁶ The alignment results show the nucleotide consensus and occupancy level within these 7 species, suggesting a high degree of sequence conservation within the 5'CL region of enteroviral genomes.

Supplementary Figure 20. The R-Scape (RNA Structural Covariation Above Phylogenetic Expectation) analysis (<http://eddylab.org/R-scape/>)²⁷ of the enteroviral 5'CL consensus secondary structure. As shown in Supplementary Figure 19 above, the 5'CL sequence alignments for all 7 enteroviral species were further refined to remove duplicate sequences using FASTX-Toolkit (http://hannonlab.cshl.edu/fastx_toolkit/), and the top 1000 most complete sequences were aligned via MuscleWS (<http://www.compbio.dundee.ac.uk/jabaws/>).²⁵ These aligned sequences were then uploaded into the R-Scape²⁷ to compute a consensus secondary structure of the enteroviral 5'CL. The secondary structure analysis indicates that the sequence and the structural features within each subdomain sA, sB, sC, and sD are highly conserved among known enteroviral species.

Lines 493-494: The assertion that the lengths of the P3 helix and P4 helix are conserved is unsupported. Please include a supplementary figure to rigorously support this.

Authors: Based on the new bioinformatic analysis as suggested by the reviewer (discussed above), we have included a new Supplementary Figure 21 to support our discussion about the length of the P3 and P4 helix (also shown below). The length P4 helix preceding the Py-Py region also supports the requirement for the P3 helix length to be conserved, which is needed to precisely position the A40 for interactions with the central Py-Py C56•U73 pair within the P4 helix.

NC_0099961_Human_rhinovirus_C
 NC_0383111_Human_rhinovirus_l_strain_AUC VR-1559
 NC_0016171_Human_rhinovirus_89
 NC_0389891_Picornaviridae_sp_rodent
 NC_0014901_Rhinovirus_B14_complete_sequence
 NC_0383121_Human_rhinovirus_3
 NC_0383091_Simian_enterovirus_SV4_strain_1715_UWB
 NC_0342671_Enterovirus_goat_JL14
 NC_0383101_Dromedary_camel_enterovirus_strain_19Cc
 NC_0018591_Bovine_enterovirus
 NC_0087141_Possum_enterovirus_W1
 NC_0298541_Yak_enterovirus_strain_SWU N-AB001
 NC_0336951_Enterovirus_AN12_genomic_RNA_strain_AN12
 NC_0212201_Enterovirus_F_strain_BEV-261
 NC_0020583_Poliovirus
 NC_0383061_Human_coxsackievirus_A2_strain_Fleetwood
 NC_0016121_Human_enterovirus_A
 NC_0304541_Enterovirus_A114_strain_V13-0285
 NC_0014301_Human_enterovirus_D
 NC_0383081_Human_enterovirus_68_strain_Fermon
 NC_0383071_Coxsackievirus_B3_mRNA
 NC_0014721_Human_enterovirus_B
 NC_0240731_Enterovirus_sp_isolate_CPM L 8109_08
 NC_0136951_Enterovirus_J_strain_N203
 NC_0104151_Enterovirus_J_strain_1631

Supplementary Figure 21. The sequence alignment and base-pair conservation for the enteroviral 5'CL sequences. The alignment of the 25 reference enteroviral sequences available in the NCBI RefSeq database (<https://www.ncbi.nlm.nih.gov/refseq/>) with complete 5'CL sequences was performed using NCBI Nucleotide BLAST tool (<https://blast.ncbi.nlm.nih.gov/Blast.cgi>). The resulting multi-alignment file was visualized using the R-Chie (RNA Arc Diagram web server, <https://www.e-rna.org/r-chie/>).²⁸ The results further elucidate the high conservation of the base pair numbers within 5'CL subdomain helices and their alignment among all members of the enterovirus genus. Specifically, the length of the sC helix (4 base pairs) and sD helix preceding the Py-Py region (6 base pairs interrupted by dinucleotide bulge) appear highly conserved, which is consistent with the viral requirement for maintaining the long-range interactions between the sC A40 and the central C•U pair in the sD Py-Py helix as observed in our CVB3 5'CL crystal structure.

Response to Reviewers References

1. Das, R., Karanicolas, J. & Baker, D. Atomic accuracy in predicting and designing noncanonical RNA structure. *Nature methods* **7**, 291 (2010).
2. Lyskov, S. et al. Serverification of Molecular Modeling Applications: The Rosetta Online Server That Includes Everyone (ROSIE). *PLOS ONE* **8**, e63906 (2013).
3. Du, Z., Yu, J., Andino, R. & James, T.L. Extending the Family of UNCG-like Tetraloop Motifs: NMR Structure of a CACG Tetraloop from Coxsackievirus B3. *Biochemistry* **42**, 4373-4383 (2003).
4. Du, Z., Yu, J., Ulyanov, N.B., Andino, R. & James, T.L. Solution structure of a consensus stem-loop D RNA domain that plays important roles in regulating translation and replication in enteroviruses and rhinoviruses. *Biochemistry* **43**, 11959-72 (2004).
5. Ohlenschläger, O. et al. The structure of the stemloop D subdomain of coxsackievirus B3 cloverleaf RNA and its interaction with the proteinase 3C. *Structure* **12**, 237-48 (2004).
6. Tolbert, B.S. et al. Major groove width variations in RNA structures determined by NMR and impact of ¹³C residual chemical shift anisotropy and ¹H-¹³C residual dipolar coupling on refinement. *J Biomol NMR* **47**, 205-19 (2010).
7. Murray, K.E. & Barton, D.J. Poliovirus CRE-dependent VPg uridylylation is required for positive-strand RNA synthesis but not for negative-strand RNA synthesis. *Journal of virology* **77**, 4739-4750 (2003).
8. van Ooij, M.J. et al. Structural and functional characterization of the coxsackievirus B3 CRE (2C): role of CRE (2C) in negative-and positive-strand RNA synthesis. *Journal of general virology* **87**, 103-113 (2006).
9. Goodfellow, I. et al. Identification of a cis-acting replication element within the poliovirus coding region. *Journal of virology* **74**, 4590-4600 (2000).
10. Rieder, E., Paul, A.V., Kim, D.W., van Boom, J.H. & Wimmer, E. Genetic and biochemical studies of poliovirus cis-acting replication element cre in relation to VPg uridylylation. *Journal of virology* **74**, 10371-10380 (2000).
11. Morasco, B.J., Sharma, N., Parilla, J. & Flanagan, J.B. Poliovirus cre (2C)-dependent synthesis of VPgUpU is required for positive-but not negative-strand RNA synthesis. *Journal of virology* **77**, 5136-5144 (2003).
12. Barton, D.J., O'Donnell, B.J. & Flanagan, J.B. 5' cloverleaf in poliovirus RNA is a cis-acting replication element required for negative-strand synthesis. *The EMBO journal* **20**, 1439-1448 (2001).
13. Sharma, N. et al. Functional role of the 5' terminal cloverleaf in Coxsackievirus RNA replication. *Virology* **393**, 238-249 (2009).
14. Brown, D.M., Cornell, C.T., Tran, G.P., Nguyen, J.H.C. & Semler, B.L. An Authentic 3' Noncoding Region Is Necessary for Efficient Poliovirus Replication. *Journal of Virology* **79**, 11962-11973 (2005).
15. Zell, R., Sidigi, K., Bucci, E., Stelzner, A. & Görlach, M. Determinants of the recognition of enteroviral cloverleaf RNA by coxsackievirus B3 proteinase 3C. *RNA (New York, N. Y.)* **8**, 188-201 (2002).
16. Bailey, J.M. & Tapprich, W.E. Structure of the 5' nontranslated region of the coxsackievirus B3 genome: Chemical modification and comparative sequence analysis. *J Virol* **81**, 650-68 (2007).
17. Prusa, J., Missak, J., Kittrell, J., Evans, J.J. & Tapprich, W.E. Major alteration in coxsackievirus B3 genomic RNA structure distinguishes a virulent strain from an avirulent strain. *Nucleic Acids Research* **42**, 10112-10121 (2014).
18. Mahmud, B., Horn, C.M. & Tapprich, W.E. Structure of the 5' Untranslated Region of Enteroviral Genomic RNA. *J Virol* **93**(2019).

19. Warden, M.S. et al. Conformational flexibility in the enterovirus RNA replication platform. *RNA* **25**, 376-387 (2019).
20. Roehl, H.H. & Semler, B.L. Poliovirus infection enhances the formation of two ribonucleoprotein complexes at the 3' end of viral negative-strand RNA. *Journal of Virology* **69**, 2954-2961 (1995).
21. Banerjee, R., Echeverri, A. & Dasgupta, A. Poliovirus-encoded 2C polypeptide specifically binds to the 3'-terminal sequences of viral negative-strand RNA. *Journal of Virology* **71**, 9570-9578 (1997).
22. Banerjee, R., Tsai, W., Kim, W. & Dasgupta, A. Interaction of Poliovirus-Encoded 2C/2BC Polypeptides with the 3' Terminus Negative-Strand Cloverleaf Requires an Intact Stem-Loop b. *Virology* **280**, 41-51 (2001).
23. Brunner, J.E. et al. Functional Interaction of Heterogeneous Nuclear Ribonucleoprotein C with Poliovirus RNA Synthesis Initiation Complexes. *Journal of Virology* **79**, 3254-3266 (2005).
24. Ertel, K.J., Brunner, J.E. & Semler, B.L. Mechanistic consequences of hnRNP C binding to both RNA termini of poliovirus negative-strand RNA intermediates. *J Virol* **84**, 4229-42 (2010).
25. Edgar, R.C. Muscle5: High-accuracy alignment ensembles enable unbiased assessments of sequence homology and phylogeny. *Nature Communications* **13**, 6968 (2022).
26. Waterhouse, A.M., Procter, J.B., Martin, D.M.A., Clamp, M. & Barton, G.J. Jalview Version 2—a multiple sequence alignment editor and analysis workbench. *Bioinformatics* **25**, 1189-1191 (2009).
27. Rivas, E., Clements, J. & Eddy, S.R. A statistical test for conserved RNA structure shows lack of evidence for structure in lncRNAs. *Nature Methods* **14**, 45-48 (2017).
28. Lai, D., Procter, J.R., Zhu, J.Y.A. & Meyer, I.M. R-chie : a web server and R package for visualizing RNA secondary structures. *Nucleic Acids Research* **40**, e95-e95 (2012).

REVIEWERS' COMMENTS

Reviewer #1 (Remarks to the Author):

This work is clearly worthwhile to publish. My main remaining concern is lack of concrete evidence for the loop to bulge contact from NMR data, but the circumstantial evidence provided is not insubstantial.

Reviewer #2 (Remarks to the Author):

In this manuscript, Das and co-workers present a high-resolution crystal structure of the 5' cloverleaf sequence from coxsackievirus B3, an important enterovirus that also serves as a model system for several other medically relevant viruses. The 5' CL is an essential RNA structure required for replication and pathogenicity across the enterovirus family. The authors used a strategy of modifying the 5'CL sequence to bind a Fab fragment to favor crystallization. Using these alterations to the 5'CL they were able to determine the 1.9 angstrom crystal structure of the modified RNA. Importantly, they demonstrate that the inclusion of the Fab binding site does not alter the interactions of the RNA with its 3C binding partner. Another single point mutation was used to increase the resolution of this structure. An analysis of the location of this mutation suggests that it is unlikely that this would alter the global conformation of this RNA.

The quality of the structural biology in this work is high and the 1.9 angstrom resolution permits the observation of several interesting features of the 5'CL. One striking observation is that the 5'CL RNA in fact forms an H-type four-way junction structure with two coaxially stacked helices, a novel and interesting finding. Additional features include a triloop, a bulge sequence, and a region of unusual pyrimidine-pyrimidine base pairing. The authors determine that a hyper-conserved adenine residue within the triloop forms a long-range interaction with the py-py region, supporting the formation of the four-way junction. This provides an explanation for the conservation of these features with the 5'CL. The authors also identify a base triple interaction in a bulge sequence. Further investigation of this bulge determines that it contributes to the protein binding activity of the RNA.

The authors support their structural biology work with biochemical experiments, probing the contributions of different structural elements towards binding of the viral 3C protease and host poly(C)-binding protein. Their work points to the four-way junction as an element which organizes the two binding regions on distal sides of the RNA permitting simultaneous binding. There are interesting nuances in this work that deserves further exploration (one interesting alternative is that RNA dynamics

play an important regulatory role in the structure), however these are outside the scope of this current work.

The authors place their study in the context of previous structural and biochemical work which overall agree with their conclusions. They also present a comprehensive enterovirus 5'CL structural alignment demonstrating the conservation of the features observed in their structure.

Overall, this study represents an important advance in the field, with high-quality structural biology work and appropriate biochemical studies. The authors have sufficiently addressed my comments from the first round of review and I would recommend the acceptance of the manuscript.

A point-by-point response to the reviewers' comments

Reviewer #1 (Remarks to the Author)

This work is clearly worthwhile to publish. My main remaining concern is the lack of concrete evidence for the loop to bulge contact from NMR data, but the circumstantial evidence provided is not insubstantial.

Authors: We thank the reviewer for the positive feedback and constructive suggestions during the previous round of review, which helped us improve the quality of the manuscript significantly. We are studying both the crystal and cryo-EM structures of the 5'CL-Fab complexes from different enteroviral species to better understand the dynamics of the sC loop and Py-Py interactions and how such interactions stabilize and regulate the overall architecture of enteroviral 5'CL. We will report these results in a separate publication.

Reviewer #2 (Remarks to the Author)

In this manuscript, Das and co-workers present a high-resolution crystal structure of the 5' cloverleaf sequence from coxsackievirus B3, an important enterovirus that also serves as a model system for several other medically relevant viruses. The 5' CL is an essential RNA structure required for replication and pathogenicity across the enterovirus family. The authors used a strategy of modifying the 5'CL sequence to bind a Fab fragment to favor crystallization. Using these alterations to the 5'CL, they were able to determine the 1.9-angstrom crystal structure of the modified RNA. Importantly, they demonstrate that including the Fab binding site does not alter the interactions of the RNA with its 3C binding partner. Another single-point mutation was used to increase the resolution of this structure. An analysis of the location of this mutation suggests that it is unlikely that this would alter the global conformation of this RNA.

The quality of the structural biology in this work is high, and the 1.9-angstrom resolution permits the observation of several interesting features of the 5'CL. One striking observation is that the 5'CL RNA, in fact, forms an H-type four-way junction structure with two coaxially stacked helices, a novel and interesting finding. Additional features include a triloop, a bulge sequence, and a region of unusual pyrimidine-pyrimidine base pairing. The authors determine that a hyper-conserved adenine residue within the triloop forms a long-range interaction with the Py-Py region, supporting the formation of the four-way junction. This provides an explanation for the conservation of these features with the 5'CL. The authors also identify a base triple interaction in a bulge sequence. Further investigation of this bulge determines that it contributes to the protein binding activity of the RNA.

The authors support their structural biology work with biochemical experiments, probing the contributions of different structural elements towards the binding of the viral 3C protease and host poly(C)-binding protein. Their work points to the four-way junction as an element that organizes the two binding regions on distal sides of the RNA, permitting simultaneous binding. There are

interesting nuances in this work that deserves further exploration (one interesting alternative is that RNA dynamics play an important regulatory role in the structure); however, these are outside the scope of this current work.

The authors place their study in the context of previous structural and biochemical work, which overall agrees with their conclusions. They also present a comprehensive enterovirus 5'CL structural alignment demonstrating the conservation of the features observed in their structure.

Overall, this study represents an important advance in the field, with high-quality structural biology work and appropriate biochemical studies. The authors have sufficiently addressed my comments from the first round of review, and I would recommend the acceptance of the manuscript.

Authors: We thank the reviewer for the critical comments and constructive suggestions, which helped us significantly improve the manuscript's quality from the original submission. We appreciate the reviewer's advice about studying how RNA dynamics play an important regulatory role in the context of 5'CL structure. We are investigating the crystal and cryo-EM structures of the 5'CL-Fab complexes from different enteroviral species and how these structures interact with human PCBP2 protein. These studies will be published separately.